# GAUSS: Graph-Assisted Uncertainty Quantification using Structure and Semantics for Long-Form Generation in LLMs

## Abstract

In high-stakes domains such as clinical reporting, legal analysis, and policy drafting, large language models (LLMs) are increasingly tasked with generating extended, fact-rich narratives rather than isolated sentences. Accurately quantifying uncertainty in these long-form outputs is essential for ensuring their reliability. Prior approaches either assign a single confidence score to an entire paragraph, often using other LLMs or assess factual consistency by comparing discrete atomic facts derived from the paragraphs across multiple generations. Some recent methods also incorporate graph-based representations, modeling fact–paragraph structures as bipartite entailment graphs and derive uncertainty from node centrality of the facts. However, these methods overlook the interdependencies among atomic facts within a paragraph, as well as the explicit organizational, structural and semantic variation across multiple paragraphs generated by an LLM for the same query, thereby missing a key source of uncertainty inherent specifically to long-form generation. In this work, we introduce GAUSS (**G**raph-**A**ssisted **U**ncertainty Quantification using **S**tructure and **S**emantics for Long-Form Generation in LLMs), a principled framework for measuring uncertainty in long-form LLM outputs through graph-based alignment. Each generated paragraph is modeled as a semantic graph, where nodes represent atomic facts about the paragraph and edges capture inter-fact relationships. We hypothesize that uncertainty arises from structural and semantic discrepancies among these graphs across different generated paragraph samples. GAUSS formalizes this intuition by computing an uncertainty score as the expected alignment cost between the semantic graph of an anchor paragraph and those of alternative reference paragraphs generated by the LLM. By jointly capturing both semantic content and structural coherence of the generated texts, GAUSSmoves beyond coarse sentence-level scores to offer a more interpretable and theoretically grounded approach to uncertainty quantification. Our code is available at https://github.com/sourceuser-1/Code.

## 1 Introduction

Large Language Models (LLMs) are increasingly used in high-stakes domains requiring strong factual accuracy [25, 10, 11, 12, 8]. While they excel at fluent long-form generation, their outputs often exhibit hallucinations and inconsistencies. This makes reliable uncertainty quantification essential in critical settings. While recent advances in uncertainty quantification (UQ) have made progress on short-form generation using semantic features, conformal calibration, and entropy-based metrics [9, 13, 22, 4, 23, 7, 16, 17], these methods remain constrained to isolated sentences or atomic facts. Long-form generation, as produced by LLMs is inherently paragraphic in nature: it weaves together multiple atomic facts in a structured, interdependent manner. These facts are not merely co-located; they exhibit logical flow, hierarchical relationships, and latent semantic dependencies. Traditional UQ techniques, which assess uncertainty in isolation or via entropy over discrete outputs [16, 17], struggle to capture such organization. Recent work [31, 14, 32] has sought to extend UQ to long-form generation by decomposing paragraphs into atomic facts, evaluating each via entailment models or consistency checks, and aggregating the results into a paragraph-level score. However, treating facts as independent units ignores the structural coherence that underpins long-form content. Logical flow, contextual dependencies, and the nuanced arrangement of facts are discarded, leading to coarse and potentially misleading uncertainty estimates. Reliable, interpretable UQ for long-form generation

demands moving beyond 'bag-of-facts' analyses toward representations that reflect the structure and semantics of entire paragraphs.

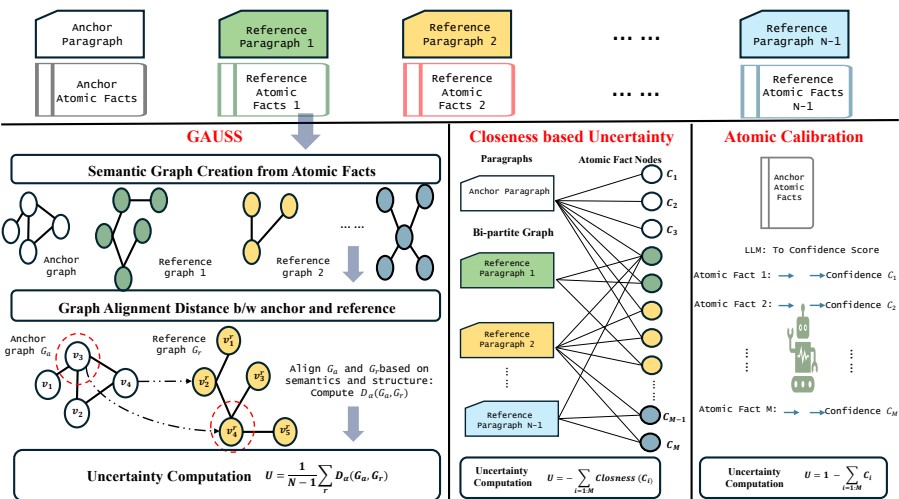

Figure 1: GAUSS decomposes paragraphs into atomic facts and represents each as a semantic graph, capturing both factual semantics and inter-fact relations. Uncertainty is computed via graph alignment between an anchor and the remaining reference graphs. In contrast, [15] uses a single bipartite graph and centrality over fact–paragraph entailments, while [31] relies on external LLMs and ignores intra-paragraph structure.

To build reliable and interpretable UQ for long-form generation, we argue that uncertainty should be grounded in both the internal organization and semantic meaning of the paragraphs themselves. In this regard, we propose GAUSS (**G**raph-**A**ssisted **U**ncertainty quantification using **S**tructure and **S**emantics for Long Form Generations in LLMs), a principled framework for modeling uncertainty in long-form LLM outputs through graph-based representations (see Figure 1). In GAUSS, each generated paragraph is first decomposed into its constituent atomic facts, which are represented as nodes in a semantic graph. Edges between these nodes encode pairwise semantic relationships, capturing dependencies among factual elements. This graph-based abstraction serves two key roles: (i) it preserves the symbolic structure inherent in long-form text, and (ii) it embeds the semantic content of individual atomic facts. To quantify uncertainty, GAUSS compares the semantic graph of a generated paragraph against those of other candidate generations, measuring structural and semantic deviations using a graph alignment distance. In doing so, GAUSS offers a structure-aware approach to UQ in long form generation. A recent graph-based approach [15] represents atomic fact–paragraph interactions using a single bipartite entailment graph and quantifies fact-level uncertainty via graph centrality metrics such as closeness (see Figure 1). [15] model fact level uncertainty through relationships between facts and paragraphs across generations, without explicitly capturing the rich internal organization/ coherence of atomic facts in individual paragraphs. In contrast, GAUSS takes a fundamentally different approach: it constructs a separate semantic graph for each paragraph, with nodes representing atomic facts and edges encoding semantic and structural dependencies. This per-paragraph semantic graph modeling enables GAUSS to assess uncertainty at generated paragraph level by directly comparing structure and meaning across generations. Furthermore, unlike [15], GAUSS offers theoretical guarantees, and extends naturally to atomic-level uncertainty (GAUSS-atomic), content filtering applications etc (Appendix Sections 2-3). [24, 5] leverage graph structures for reasoning-based uncertainty, using deducibility or explanation graphs with conformal prediction or graph-edit distances. GAUSS instead targets long-form outputs by aligning semantic graphs of atomic facts. Thus, our key contributions are:

1. We introduce a semantic-graph representation that simultaneously encodes the meaning of each atomic fact and the relational dependencies among them to capture both content and structure in long-form paragraphs.

2. We propose GAUSS, a structure and semantics-aware framework for uncertainty quantification in long-form generation, which estimates uncertainty via fused Gromov–Wasserstein graph alignment distance between semantic graphs.

3. We theoretically establish the Lipschitz continuity of both the graph alignment distance and the resulting uncertainty measure under semantic and structural perturbations, ensuring robustness to small graph structure and semantic embedding variations in the generated text.

4. We derive exponential convergence bounds for the uncertainty measure in terms of the number of sampled paragraphs and the consistency of the generating LLM, demonstrating that reliable uncertainty estimates can be obtained with modest sample sizes.

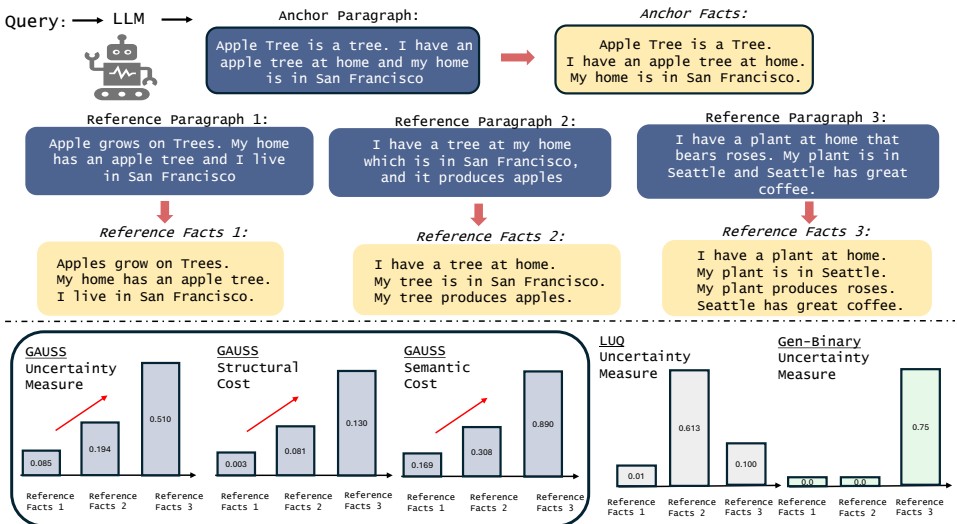

Figure 2: An anchor paragraph and three references diverge in semantics and structure. GAUSS uses graph-based structural + semantic costs to produce steadily increasing uncertainty (left), whereas semantic-only baselines (right) give inconsistent scores, highlighting the need for joint structure–semantics comparison.

## 2 MOTIVATION

We posit that uncertainty in long-form generation should reflect the variability in how atomic factual units (atomic facts) within the generated paragraphs are represented and semantically interconnected. To operationalize this concept, consider a scenario wherein an LLM generates multiple paragraphs in response to the same query for a long-form answer. We can designate one paragraph as the anchor and all others as reference paragraphs and decompose each generated paragraph into its constituent atomic facts. Consequently, uncertainty in paragraph generation can be assessed by examining the structural and semantic discrepancies between the atomic facts of the anchor and reference paragraphs.

To analyze such differences rigorously, we represent each paragraph as a semantic graph, where nodes encode the meaning of individual atomic facts, while edges quantify semantic relationships between them. This structured representation enables us to capture not only the content of a paragraph but also the relational fabric that holds it together. By comparing these graphs across generations, we obtain an interpretable measure of uncertainty, one that reflects both the semantic variation and organizational shifts inherent to long-form language generation.

Figure 2 shows three reference paragraphs that have been constructed so as to gradually diverge from an anchor in both meaning and structure. Methods like LUQ [32] and Gen-Binary [31] treat long-form uncertainty purely as semantic agreement : either by averaging entailment scores over atomic facts or by querying another LLM for fact support (between anchor facts and other reference paragraphs), thus overlooking how the paragraphs' symbolic structure interacts with their content. In contrast, GAUSS encodes each paragraph as a graph, jointly capturing the inherent structure (atomic fact interdependices) and semantics (atomic fact semantics). By examining the three scenarios in Figure 2, we highlight how this graph-based comparison more faithfully quantifies uncertainty.

- **Reference Facts 1:** The reference facts closely mirror the anchor facts with minor wording differences. Here, our approach correctly identifies slight structural and semantic varia-

tions, consistently demonstrated across comparative methods (GAUSS , Gen-Binary[31], LUQ[32]).

- **Reference Facts 2:** These facts retain overall semantic congruence with the anchor, but differ significantly in their organization, lacking direct one-to-one correspondence with the anchor facts. Unlike approaches relying solely on aggregate semantic similarity, GAUSS explicitly captures variations in generation style and structure, yielding a higher uncertainty score. Conversely, Gen-Binary underestimates uncertainty and fails to recognize different organization style as it solely relies on just the overall semantics.

- **Reference Facts 3:** This scenario introduces significant alterations in both semantics and structure (through contradictory information and extraneous content) in comparison to the anchor facts. Our method accurately identifies these substantial deviations, reflecting increased structural and semantic costs and thereby a higher uncertainty measure.

This example demonstrates our method's sensitivity to nuanced variations in paragraph semantics, structure, and style. In contrast, Gen-Binary [31], which focuses solely on overall semantics, fails to detect organizational differences, overlooking uncertainties evident in reference facts 2. Similarly, LUQ [32], by disregarding structural organization, does not accurately capture the progression of uncertainty across reference facts. Unlike these methods, GAUSS consistently reflects a monotonic increase in uncertainty across the progressively varying reference paragraphs, emphasizing the need for a structure and semantic aware uncertainty quantification technique for long-form LLM generation.

## 3 BACKGROUND

Although it has not yet been applied to uncertainty quantification in long-form generations by LLMs, graph-based optimal transport (OT) [18, 28] offers a principled framework for comparing structured data such as graphs, by aligning the nodes of two graphs based on both structural relationships and feature-level similarity. While classical OT aligns distributions over Euclidean spaces, graph alignment requires accounting for relational dependencies between nodes.

Let $G_1 = (V_1, E_1, C_1, \ell_f)$ and $G_2 = (V_2, E_2, C_2, \ell_f)$ be two graphs, where $V_i$ and $E_i$ represent the nodes and edges of the graphs respectively. $C_1 \in \mathbb{R}^{|V_1| \times |V_1|}$ and $C_2 \in \mathbb{R}^{|V_2| \times |V_2|}$ are structure matrices encoding pairwise node relations (e.g., shortest path, adjacency etc), and $\ell_f$ is the node feature mapping function. The goal in graph based OT is to align the two graphs $G_1$ and $G_2$ based on both the structural and semantic similarity. The *fused Gromov–Wasserstein distance* (graph alignment distance) $D_\alpha$ [28, 30], compares both structure and features via:

$$D_\alpha(G_1, G_2) = \min_{\pi \in \Pi} \sum_{i,j,k,\ell} \left[ (1-\alpha) \underbrace{m\big(\ell_f(i), \ell_f(j)\big)}_{\text{semantic cost}} + \alpha \underbrace{|C_1(i,k) - C_2(j,\ell)|}_{\text{structural cost}} \right] \pi_{ij}\, \pi_{k\ell}\,,$$

$$\text{subject to} \quad \Pi = \left\{ \pi \in [0,1]^{|V_1| \times |V_2|} \;\middle|\; \sum_{j=1}^{|V_2|} \pi_{ij} = 1, \; \sum_{i=1}^{|V_1|} \pi_{ij} = 1 \right\}$$

(3.1)

where $m(\cdot, \cdot)$ is a feature cost function (e.g., cosine distance), and $\alpha \in [0, 1]$ trades off structural and feature alignment. The coupling $\pi$ can be seen as the mapping from the nodes of $G_1$ to nodes of $G_2$.

## 4 GRAPH BASED UNCERTAINTY QUANTIFICATION FOR LONG FORM GENERATION

### 4.1 OVERVIEW OF THE LONG FORM GENERATION UNCERTAINTY ESTIMATION FRAMEWORK

We begin by outlining the core pipeline of GAUSS before detailing each component of the framework.

To quantify the long-form uncertainty associated with a given query $q$ and language model $\mathcal{M}$, we first sample $N$ independent paragraph-length responses:

$$\{P_i\}_{i=1}^N \sim \mathcal{M}(q).$$

From these, we designate one paragraph $P_a$ as the *anchor*, and treat the remaining $N - 1$ paragraphs $\{P_r\}_{r \neq a}$ as *references*. Each paragraph $P_i$ is decomposed into a set of atomic facts $\mathcal{F}_{P_i}$ using a factual decomposition model $\mathcal{M}_{\text{atomic}}$, and is subsequently represented as a semantic graph $G_i = (V_i, E_i, C_i, \ell_f)$, where nodes encode the atomic facts and edges capture semantic dependencies.

To assess the variability of generations around the anchor, we compute the alignment distance $D_\alpha(G_a, G_r)$ between the anchor graph $G_a$ and each reference graph $G_r$. The final uncertainty score for query $q$ is given by the mean alignment cost:

$$U(q) = \frac{1}{N - 1} \sum_{r \neq a} D_\alpha(G_a, G_r). \tag{4.1}$$

Intuitively, $U(q)$ captures the average structural and semantic deviation between the anchor and reference paragraphs—higher values indicate greater generation variability and, thus, higher uncertainty. We describe the construction of semantic graphs in Section 4.2, the computation of alignment distances in Section 4.3, and the theoretical properties of the uncertainty measure in Section 4.4. A full overview of the GAUSS pipeline is provided in Algorithm 1.

## 4.2 REPRESENTING PARAGRAPHS AS SEMANTIC GRAPHS

To capture both the semantic content and the internal relational structure within a paragraph $P_i$, we represent it as a semantic graph $G_i = (V_i, E_i, C_i, \ell_f)$. The construction of the semantic graph proceeds in three stages:

**Step 1: Atomic Fact Extraction.** We begin by decomposing paragraph $P_i$ into its constituent atomic facts:

$$\mathcal{F}_{P_i} = \mathcal{M}_{\text{atomic}}(P_i) = \{f_1, f_2, \ldots, f_{n_i}\},$$

using a prompted model $\mathcal{M}_{\text{atomic}}$ following prior work [29, 20, 31, 32]. Each $f_k$ denotes a standalone factual statement derived from $P_i$. Further implementation details for prompting $\mathcal{M}_{\text{atomic}}$ are provided in the Appendix.

**Step 2: Node Construction and Semantic Embedding.** Each atomic fact $f_k \in \mathcal{F}_{P_i}$ is treated as a node $v_k$ in the vertex set $V_i$, so that $V_i = \{v_1, \ldots, v_{n_i}\}$. To encode the semantic meaning of each node, we use a sentence embedding model $\mathcal{M}_{\text{sentence}}$ to compute:

$$\ell_f(v_k) = \mathcal{M}_{\text{sentence}}(f_k) \in \mathbb{R}^d.$$

**Step 3: Structural Matrix Construction.** To encode inter-fact dependencies, we define the structure matrix $C_i \in \mathbb{R}^{n_i \times n_i}$ using pairwise semantic distance:

$$C_i(k, \ell) = 1 - \cos\left(\ell_f(v_k), \ell_f(v_\ell)\right),$$

where $\cos(\cdot, \cdot)$ denotes cosine similarity between embeddings. Higher values in $C_i(k, \ell)$ correspond to a weaker semantic affinity between facts $f_k$ and $f_\ell$. While we adopt semantic distance here, other graph-based metrics such as graph kernels may also be more generally used. We explore such variants in Section 4 of the Appendix.

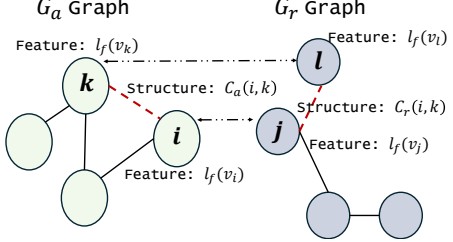

Figure 3: The structural and feature cost while aligning graph $G_a$ and reference graph $G_r$.

The resulting graph $G_i = (V_i, E_i, C_i, \ell_f)$ serves as a compact, interpretable representation of $P_i$ and captures both the meaning of each atomic fact and their mutual relationships, setting the stage for principled alignment across paragraph representations.

## 4.3 COMPUTING ALIGNMENT DISTANCE BETWEEN SEMANTIC GRAPHS

In this subsection, we detail the computation of the alignment distance $D_\alpha(G_a, G_r)$ between the anchor graph $G_a$ and a reference graph $G_r$, with $n_a$ and $n_r$ nodes respectively. The process involves constructing the semantic cost matrix $M^r$ and the structural cost tensor $L^r$, which jointly define the alignment objective. An illustration is provided in Figure 3.

**Semantic Cost Matrix $M^r$.** The semantic cost matrix $M^r \in \mathbb{R}^{n_a \times n_r}$ encodes the dissimilarity at the feature level between the atomic facts in the anchor graph $G_a$ and those in the reference graph $G_r$. For each pair of nodes $v_i \in G_a$ and $v_j \in G_r$, the cost entry $M^r[i,j]$ is defined as:

$$M^r[i,j] = 1 - \cos\big(\ell_f(v_i), \ell_f(v_j)\big) \cdot \mathcal{M}_{\text{entail}}\big(\ell_f(v_i), P_r\big), \tag{4.2}$$

where $\mathcal{M}_{\text{entail}}(\ell_f(v_i), P_r) \in \{0, 1\}$ is the output of model that determines whether the atomic fact $\ell_f(v_i)$ is supported by the reference paragraph $P_r$. Implementation details and prompting strategies for $\mathcal{M}_{\text{entail}}$ are provided in the Appendix.

This formulation combines both local and global signals: the cosine similarity reflects the local semantic alignment between atomic facts, while the entailment signal serves as a global gating mechanism—suppressing alignment when the reference paragraph lacks the corresponding factual support. Together, they ensure that only semantically and contextually consistent node alignments are favored in the downstream graph matching process.

**Structural Cost Tensor $L^r$.** The structural cost tensor $L^r \in \mathbb{R}^{n_a \times n_a \times n_r \times n_r}$ measures alignment consistency between the internal topologies of the two semantic graphs. Recall that each structure matrix $C_a$ or $C_r$ is computed as:

$$C(i,k) = 1 - \cos\big(\ell_f(v_i), \ell_f(v_k)\big),$$

which reflects the semantic dissimilarity between atomic fact embeddings within a graph. Given this, we define the structural cost tensor as:

$$L^r[i,j,k,\ell] = |\, C_a(i,k) - C_r(j,\ell) \,|, \tag{4.3}$$

which penalizes mismatches in the relative connectivity of node pairs $(i,k)$ in $G_a$ and $(j,\ell)$ in $G_r$.

**Final Alignment Distance.** With the semantic cost matrix $M^r$ and the structural cost tensor $L^r$ defined, the alignment distance between $G_a$ and $G_r$, $\mathrm{D}_\alpha(G_a, G_r)$ is computed as:

$$\mathrm{D}_\alpha(G_a, G_r) = \min_{\pi \in \Pi} \sum_{i,j,k,\ell} \Big[ (1-\alpha)\, M^r[i,j] + \alpha\, L^r[i,k,j,\ell] \Big] \pi_{ij} \pi_{k\ell},$$

where $\pi$ is a stochastic map which aligns nodes in $G_a$ to nodes in $G_r$. This alignment distance reflects the closest possible semantic similarity and global structural coherence between the anchor and reference graphs through the mapping $\pi$.

---

**Algorithm 1** GAUSS

---

**Require:** Query $q$, language model $\mathcal{M}$, number of samples $N$, trade-off parameter $\alpha$
 1: **Sample Paragraphs:** Generate $N$ long-form responses $\{P_i\}_{i=1}^N \sim \mathcal{M}(q)$
 2: **Select Anchor:** Choose one sample $P_a$ as the anchor paragraph; set remaining $\{P_r\}_{r \neq a}$ as reference paragraphs
 3: **for** each paragraph $P_i$ **do**
 4:     Extract atomic facts $\mathcal{F}_{P_i}$
 5:     Construct semantic graph $G_i = (V_i, E_i, C_i, \ell_f)$
 6: **end for**
 7: **for** each reference graph $G_r$, where $r \neq a$ **do**
 8:     Compute alignment distance $\mathrm{D}_\alpha(G_a, G_r)$
 9: **end for**
10: **Expectation:** Compute average alignment cost $U(q) = \frac{1}{N-1} \sum_{r \neq a} \mathrm{D}_\alpha(G_a, G_r)$
11: **return** $U(q)$

---

### 4.4 Theoretical Properties of the Proposed Uncertainty Measure

In this section, we establish theoretical guarantees for the proposed uncertainty measure $U(q)$, defined in equation 4.1 as the mean alignment cost between an anchor graph $G_a$ and reference graphs $\{G_r\}_{r \neq a}$ via the distance $\mathrm{D}_\alpha$. We first prove the Lipschitz continuity of $\mathrm{D}_\alpha$ in Lemma 4.1, and extend it to $U(q)$ in Theorem 4.1, ensuring robustness to semantic and structural perturbations. Theorem 4.2 further establishes exponential convergence of $U(q)$ to its expectation, governed by the number of reference samples and the LLM's consistency, thus offering practical guidance on the sampling budget required for stable uncertainty estimation.

**Remark** 1. The following results provide continuity and convergence guarantees for the uncertainty measure $U(q)$, grounded in graph representations of generated paragraphs. More generally, these properties extend to uncertainty quantification in graph generation, which we formalize below.

Suppose the anchor graph $G_a$ has $n_a$ nodes and a reference graph $G_r$ has $n_r$ nodes. We define the *feature cost matrix* $M^r \in \mathbb{R}^{n_a \times n_r}$ by $M^r[t, h] = m\big(\ell_f(t), \ell_f(h)\big)$, which measures the semantic dissimilarity between node $t \in G_a$ and node $h \in G_r$. We also define the *structural cost tensor* $L^r \in \mathbb{R}^{n_a \times n_a \times n_r \times n_r}$ as $L^r[i, k, j, \ell] = |C_a(i, k) - C_r(j, \ell)|$, where $C_a$ and $C_r$ are the structure matrices of $G_a$ and $G_r$, respectively. This tensor captures discrepancies in pairwise structural relations across the two graphs. For our application to long-form generation, the specific definitions of $M^r$ and $L^r$ are provided in Section 4.3. Together, they fully specify the inputs to the alignment distance $D_\alpha(M^r, L^r)$ used in computing uncertainty.

**Remark** 2. For any fixed pair of graphs $G_a$ and $G_r$, we use the notation $D_\alpha(G_a, G_r)$ interchangeably with $D_\alpha(M^r, L^r)$, where $M^r$ and $L^r$ are the corresponding feature cost matrix and structural cost tensor derived from $G_a$ and $G_r$. Thus $D_\alpha$ - *graph alignment distance*, can be alternatively written as:

$$D_\alpha(M^r, L^r) = \min_{\pi \in \Pi} \sum_{i,j,k,\ell} \left[ (1 - \alpha) M^r[i, j] + \alpha L^r[i, k, j, \ell] \right] \pi_{ij} \pi_{k\ell}$$

**Lemma 4.1** (Lipschitz Continuity of Alignment Distance). *The alignment distance $D_\alpha(M^r, L^r)$ is Lipschitz continuous with respect to both the feature cost matrix $M^r$ and the structural cost tensor $L^r$. That is for any $(M^r, L^r), (\widetilde{M^r}, \widetilde{L^r})$,*

$$\left| D_\alpha(M^r, L^r) - D_\alpha(\widetilde{M^r}, \widetilde{L^r}) \right| \leq (1 - \alpha)\|M^r - \widetilde{M^r}\|_\infty + \alpha\|L^r - \widetilde{L^r}\|_\infty.$$

**Theorem 4.1** (Lipschitz Continuity of the Proposed Uncertainty Measure). *Let $U(q)$ denote the uncertainty score for query $q$ with respect to a generated graph $G_a$ as the* anchor, *and $N - 1$ independently generated graphs $\{G_r\}_{r \neq a}$ as reference graphs, defined as*

$$U(q) = \frac{1}{N - 1} \sum_{r \neq a} D_\alpha(M^r, L^r), \tag{4.4}$$

*where $(M^r, L^r)$ are the semantic cost matrices and structural cost tensors between $G_a$ and $G_r$. Then $U(q)$ is Lipschitz continuous with respect to the collection $\{M^r, L^r\}_{r \neq a}$.*

**Theorem 4.2** (Convergence of the Uncertainty Measure). *Under general boundedness assumptions on the alignment distances, the uncertainty score $U(q)$ defined in Eqn (4.4) exponentially converges around the true mean $\mathbb{E}[U(q)]$, specifically:*

$$\mathbb{P}\left[ |U(q) - \mathbb{E}[U(q)]| > \epsilon \right] \leq 2\exp\left( -\frac{2(N - 1)\epsilon^2}{D^2} \right),$$

*for any $\epsilon$ and some constant $D > 0$ that depends on the graph generation inconsistency.*

**Implication of Theorems 4.1 & 4.2:** The Lipschitz property of $U(q)$ ensures that small semantic or structural perturbations in input graphs lead to proportionally bounded changes in the uncertainty score, making the uncertainty measure robust to noise in the structure extraction or feature embedding process. We provide a practical illustration of the robustness of the uncertainty measure to the embedding process in Section 5 of the Appendix. Theorem 4.2 shows that $U(q)$ concentrates exponentially around its expectation, enabling reliable uncertainty estimation from a modest number of reference samples $N - 1$. Furthermore, for LLMs with low graph generation inconsistency factor $D$, $U(q)$ converges even faster, thus, requiring fewer samples. We provide illustrations of the convergence behavior and inconsistency factor $D$, with two representative LLMs in Section 7 of the Appendix. We provide proofs of the above theorems in Section 1 of the Appendix. We also provide a discussion on the computational cost and runtime of GAUSS in Section 9 of the Appendix.

## 5 EXPERIMENTS

We evaluate our framework by comparing graph-based uncertainty to factual correctness across three benchmarks and multiple LLMs. [*Datasets*] Specifically, we assess performance on: (1) 183 biography prompts from Bios [20], verified via Wikipedia; (2) 500 open-ended queries from LongFact [29];

and (3) 500 entity-centric samples from the WildHallu chatbot corpus [33]. [***Calibration***] We follow standard practice [31, 32, 14] in evaluating uncertainty by correlating it with the truthfulness of the generated paragraph. We posit that an LLM producing structurally and semantically divergent responses for the same query is more likely to generate low-veracity content. Thus, a high uncertainty score from GAUSS should correspond to lower factuality. For each query, we compute the factuality of the anchor paragraph $P_a$ at the atomic fact level. To accomplish this, we follow the SAFE framework [29]: each atomic fact is paired with the top-ranked web search snippets given the query. Each atomic fact is then presented to the verifier model (Qwen2-32B-Instruct [3]) alongside the web search snippet. The verifier classifies whether the fact is supported, and we average these binary outcomes to yield a paragraph-level factuality score. We employ $N = 20$ paragraph generations in Algorithm 1.

Table 1: Uncertainty metrics (SC↓, PC↓, UCCE↓, QCCE↓) for different methods across three datasets.

| Method / Model | Bios | | | | LongFact | | | | WildHallu | | | |
|---|---|---|---|---|---|---|---|---|---|---|---|---|
| | SC↓ | PC↓ | UCCE↓ | QCCE↓ | SC↓ | PC↓ | UCCE↓ | QCCE↓ | SC↓ | PC↓ | UCCE↓ | QCCE↓ |
| **falcon-7b-instruct** | | | | | | | | | | | | |
| LUQ | 0.0943 | 0.0555 | 0.2261 | 0.2372 | -0.4067 | -0.4135 | 0.2865 | 0.2600 | -0.3385 | -0.2913 | 0.1295 | **0.1864** |
| Gen-Binary | -0.1535 | -0.1313 | 0.2110 | 0.1896 | -0.6443 | -0.6828 | 0.2020 | 0.2926 | **-0.7628** | **-0.7730** | 0.1654 | 0.2167 |
| Dis-Rating | 0.0134 | 0.0098 | 0.1645 | 0.2092 | -0.0479 | -0.0693 | **0.1786** | 0.3201 | 0.0418 | 0.0492 | **0.1036** | 0.3223 |
| Dis-Single | -0.1571 | -0.2625 | **0.1388** | 0.3349 | 0.0164 | 0.0991 | 0.2216 | 0.4512 | 0.0102 | 0.0240 | 0.1530 | 0.5153 |
| Centrality | -0.2670 | -0.0683 | 0.1316 | 0.1769 | -0.0795 | -0.1813 | 0.2162 | 0.3044 | -0.5441 | -0.4955 | 0.1683 | 0.2385 |
| GAUSS | **-0.4118** | **-0.3321** | 0.168 | **0.1845** | **-0.6555** | **-0.6915** | 0.1817 | **0.2199** | -0.7565 | -0.7616 | 0.1470 | 0.1978 |
| **llama3-8b-instruct** | | | | | | | | | | | | |
| LUQ | -0.0395 | -0.0546 | 0.1053 | 0.1316 | -0.0894 | -0.0452 | 0.2397 | 0.2959 | -0.4682 | -0.3939 | 0.2160 | 0.2723 |
| Gen-Binary | -0.5986 | -0.5909 | 0.1582 | 0.1256 | -0.3731 | -0.3974 | 0.1919 | 0.2726 | -0.6567 | -0.6995 | 0.2198 | 0.2740 |
| Dis-Rating | -0.6495 | -0.5372 | **0.0812** | **0.1058** | -0.2931 | -0.3204 | 0.2052 | 0.2610 | -0.6558 | -0.6779 | 0.2314 | 0.2747 |
| Dis-Single | -0.5143 | -0.5060 | 0.1156 | 0.1385 | -0.4184 | -0.3494 | 0.1889 | 0.2938 | -0.6597 | -0.6936 | 0.2704 | 0.2911 |
| Centrality | -0.6423 | -0.5429 | 0.1086 | 0.1608 | -0.3430 | -0.3379 | 0.2119 | 0.2573 | -0.5833 | -0.5971 | 0.2314 | 0.1883 |
| GAUSS | **-0.7066** | **-0.708** | 0.1035 | 0.1426 | **-0.4433** | **-0.4505** | 0.1613 | 0.2535 | **-0.6808** | **-0.7144** | 0.2048 | 0.2624 |
| **qwen2-7b-instruct** | | | | | | | | | | | | |
| LUQ | -0.0658 | -0.0698 | 0.1346 | **0.0857** | -0.2138 | -0.2378 | 0.2580 | 0.2729 | -0.3944 | -0.3830 | 0.2324 | 0.2838 |
| Gen-Binary | -0.5950 | -0.5954 | 0.1487 | 0.1232 | -0.4542 | -0.4781 | **0.1206** | 0.2732 | -0.6414 | -0.6045 | 0.2097 | 0.3080 |
| Dis-Rating | -0.4611 | -0.4535 | **0.1079** | 0.1336 | -0.3501 | -0.4355 | 0.1785 | 0.2656 | -0.5805 | -0.6726 | 0.1867 | 0.2220 |
| Dis-Single | -0.5063 | -0.5204 | 0.1144 | 0.1027 | -0.4463 | -0.4721 | 0.1695 | 0.2806 | -0.6554 | -0.6367 | 0.1826 | 0.2534 |
| Centrality | -0.6342 | -0.5023 | 0.0545 | 0.1490 | -0.4005 | -0.3164 | 0.2306 | 0.3020 | -0.5069 | -0.4956 | 0.2212 | 0.3081 |
| GAUSS | **-0.6915** | **-0.7114** | 0.1606 | 0.1505 | **-0.4979** | **-0.5233** | 0.1403 | **0.2362** | **-0.7072** | **-0.7154** | **0.1798** | **0.2212** |
| **qwen2-57b-instruct** | | | | | | | | | | | | |
| LUQ | 0.0491 | -0.0308 | 0.1263 | 0.0850 | -0.1120 | -0.0871 | 0.3133 | 0.3507 | -0.3641 | -0.3495 | 0.2900 | 0.3238 |
| Gen-Binary | -0.6093 | -0.5975 | 0.1346 | 0.0907 | -0.3328 | -0.3226 | 0.2476 | 0.3715 | -0.6434 | -0.6681 | 0.2400 | 0.2953 |
| Dis-Rating | -0.6547 | -0.5232 | **0.0676** | **0.0723** | -0.3255 | -0.3776 | 0.2038 | 0.3394 | -0.6084 | -0.6693 | 0.2603 | 0.2503 |
| Dis-Single | -0.6470 | -0.6251 | 0.1230 | 0.1732 | -0.3505 | -0.3428 | 0.1934 | 0.3340 | -0.5906 | -0.6211 | 0.2798 | 0.2154 |
| Centrality | -0.5782 | -0.5310 | 0.0844 | 0.1889 | -0.2545 | -0.1478 | 0.2015 | 0.3252 | -0.3728 | -0.4409 | 0.2448 | 0.3499 |
| GAUSS | **-0.6991** | **-0.7018** | 0.1328 | 0.1092 | **-0.4226** | **-0.4615** | 0.1844 | 0.3111 | **-0.6852** | **-0.7032** | 0.2061 | **0.2043** |
| **mistral-7b-instruct** | | | | | | | | | | | | |
| LUQ | 0.0541 | 0.0485 | 0.1302 | 0.1914 | -0.1762 | -0.1768 | 0.2548 | 0.2876 | -0.2108 | -0.1892 | 0.2946 | 0.3382 |
| Gen-Binary | -0.5803 | -0.6002 | 0.1613 | 0.1756 | -0.4138 | -0.4538 | 0.2631 | 0.2951 | -0.6889 | -0.7552 | 0.1774 | **0.2238** |
| Dis-Rating | -0.4683 | -0.4083 | 0.1539 | 0.1526 | -0.3548 | -0.3960 | 0.2511 | 0.3349 | -0.5628 | -0.6362 | **0.1200** | 0.2544 |
| Dis-Single | -0.1704 | -0.1286 | **0.0981** | 0.1769 | 0.0818 | 0.0281 | **0.1912** | 0.2880 | -0.0899 | -0.1458 | 0.1720 | 0.2970 |
| Centrality | -0.5181 | -0.5456 | 0.0421 | 0.1374 | -0.4118 | -0.3984 | 0.2679 | 0.2645 | -0.5702 | -0.5549 | 0.2750 | 0.3154 |
| GAUSS | **-0.6643** | **-0.6766** | 0.1443 | **0.1407** | **-0.4408** | **-0.4678** | 0.2525 | **0.2672** | **-0.6949** | **-0.7584** | 0.1784 | 0.2310 |

[***Baseline Methods***] We compare GAUSS against five representative baselines for long-form uncertainty estimation. Sampling-based methods generate multiple paragraphs and assess uncertainty via inter-sample disagreement. LUQ [32] (atomic variant) computes entailment probabilities between each atomic fact in the anchor and all reference paragraphs using an MNLI model; uncertainty is defined as one minus the average confidence. Gen-Binary [31] uses an LLM to assess factual support for each atomic fact across references, averaging these consistency scores to yield paragraph-level uncertainty. Single-sample methods operate on a single paragraph and query the generating LLM for internal confidence. Dis-Single [31] prompts the LLM for binary truth labels per fact, while Dis-Rating elicits 0–10 confidence scores. In both, uncertainty is computed as one minus the average per-fact confidence. However, all these baselines treat atomic facts independently, ignoring the structural and semantic relationships that GAUSS explicitly models. We also compare GAUSS with Centrality [15]. To compute the uncertainty metric from [15], we use the negative of the closeness centrality score, following the improved formulation by Wasserman and Faust. A bipartite graph is constructed over 10 generated responses and their constituent claims. To obtain a paragraph-level score, we calculate closeness for all claims within a designated anchor paragraph and take the negative average as the final uncertainty estimate. [***Evaluation Metrics***] To measure the effectiveness of the uncertainty estimates, we use multiple evaluation metrics. Spearman correlation (SC) and Pearson correlation (PC) are used

to measure monotonic and linear correlations, respectively, between uncertainty scores and factuality labels produced by [20, 29]. We also report Uniform Continuous Calibration Error (UCCE) [31] which measures the average deviation between predicted uncertainty and $1-$ ground-truth factuality across equally spaced bins: $\text{UCCE} = \sum_{m=1}^{M} \frac{|B_m|}{N} \left| \frac{1}{|B_m|} \sum_{i \in B_m} \hat{y}_i - \frac{1}{|B_m|} \sum_{i \in B_m} y_i \right|$, where $B_m$ is the $m$-th bin, $\hat{y}_i$ the normalized predicted uncertainty in the bin, and $y_i$ is the normalized $1-$ factuality score in the bin. We additionally report Quantile Continuous Calibration Error (QCCE), a variant of UCCE that uses quantile-based bins to ensure equal sample sizes. Lower UCCE and QCCE indicate better calibration between uncertainty estimates and actual truthfulness. [**_Models_**] We conduct experiments with several strong open-source Instruct LLMs, including llama3-8B [19], Mistral-7B [21], Qwen2-7B [1], Qwen2-57B [2] and Falcon-7B [27]. [**_Experimental Settings_**] We use the semantic-structural trade-off parameter $\alpha = 0.5$ in all experiments in Table 1. The sentence embedding model $\mathcal{M}_{\text{sentence}}$ used in Eqn 4.2 is mpnet-base-v2 [26]. We employ the POT library [6] to solve the $D_\alpha$ in Eqn 3.1. The $\mathcal{M}_{\text{entail}}$ used in Eqn 4.2 is Qwen2-32B-Instruct.

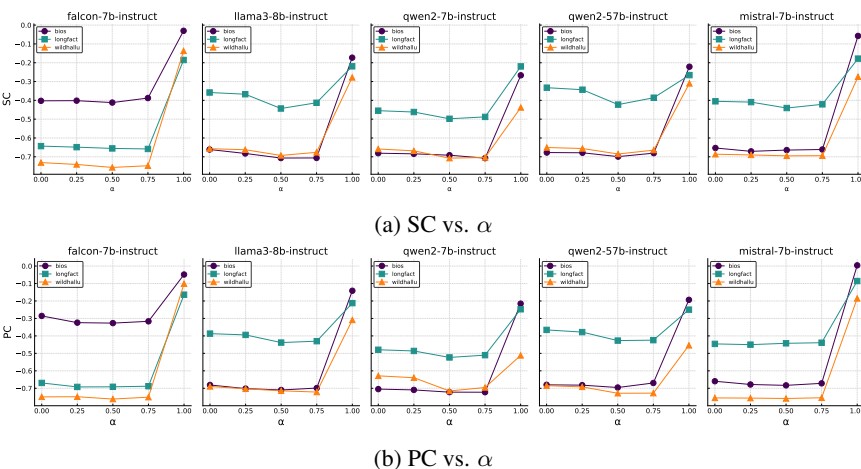

(a) SC vs. $\alpha$

(b) PC vs. $\alpha$

Figure 4: Ablation of varying the $\alpha$ parameter from 0 (only semantic cost) to 1 (only structural cost). The plots show the effect of varying $\alpha$ on the Spearman correlation (SC) and Pearson correlation (PC). The combination of both structural and semantic cost is necessary for lower SC and PC values.

Table 1 demonstrates that `GAUSS` consistently yields the strongest negative correlation with factuality across models and datasets, validating its effectiveness in capturing uncertainty. In particular, on datasets like Bios and LongFact, `GAUSS` outperforms all baselines by a significant margin in terms of the SC and PC. Baselines such as Gen-Binary, LUQ, Dis-Rating, and Dis-Single exhibit weaker and inconsistent correlation, highlighting the limitations of simple LLM inferred uncertainty metrics. The Centrality method also yields weaker correlations, underscoring the need for paragraph-level modeling of structure and semantics in uncertainty estimation across generations. Beyond correlation, `GAUSS` also maintains strong calibration performance (UCCE/QCCE), often outperforming or closely matching the best among all methods. This balance between correlation and calibration confirms the value of incorporating structural and semantic alignment via graph-based paragraph representation.

### 5.1 STRUCTURAL–SEMANTIC TRADEOFF

To assess the role of semantic and structural components in our uncertainty measure, we ablate the fusion weight $\alpha$ from 0 (semantic-only) to 1 (structural-only). As shown in Figure 4, both SC and PC correlations worsen at the extremes, with optimal performance consistently observed for intermediate $\alpha$. This confirms the complementary strengths of semantic similarity (capturing fact-level meaning) and structural alignment (capturing inter-fact relationships). Ignoring either leads to degraded performance, underscoring the importance of jointly modeling both for effective uncertainty estimation in long-form generation.

## 6 CONCLUSION

We introduce `GAUSS`, a structure- and semantics-aware framework for uncertainty quantification in long-form LLM generation. Unlike approaches that treat long-form text as a set of independent facts, `GAUSS` models both the semantic content of atomic units and their interdependencies within a paragraph to produce an uncertainty estimate. We refer the reader to the Appendix for proofs of the theorems, experimental settings etc.

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
