# APPENDIX - GAUSS: GRAPH-ASSISTED UNCERTAINTY QUANTIFICATION USING STRUCTURE AND SEMANTICS FOR LONG-FORM GENERATION IN LLMS

## 1    PROOFS OF LEMMAS AND THEOREMS

We provide the proofs of the Lemmas and Theorems below.

### 1.1    PROOF OF LEMMA 4.1

**Lemma 4.1** *(Lipschitz Continuity of Alignment Distance)* The alignment distance $D_\alpha(M^r, L^r)$ is Lipschitz continuous with respect to the feature cost matrix $M^r$ and the structural cost tensor $L^r$. That is, for any $(M^r, L^r)$ and $(\widetilde{M^r}, \widetilde{L^r})$,

$$\left| D_\alpha(M^r, L^r) - D_\alpha(\widetilde{M^r}, \widetilde{L^r}) \right| \le (1-\alpha)\|M^r - \widetilde{M^r}\|_\infty + \alpha\|L^r - \widetilde{L^r}\|_\infty.$$

**Proof.** Recall the alignment distance is defined as:

$$D_\alpha(M^r, L^r) = \min_{\pi \in \Pi} \sum_{i,j,k,\ell} \left[(1-\alpha)M^r[i,j] + \alpha L^r[i,k,j,\ell]\right] \pi_{ij}\pi_{k\ell},$$

where $\Pi = \left\{ \pi \in [0,1]^{n_a \times n_r} \;\middle|\; \sum_{j=1}^{n_r} \pi_{ij} = 1, \; \sum_{i=1}^{n_a} \pi_{ij} = 1 \right\}$ is the set of admissible couplings between nodes of the anchor graph and reference graph with $n_a$ and $n_r$ nodes respectively.

For an arbitrary coupling $\pi \in \Pi$ we define:

$$D_\alpha(M^r, L^r; \pi) = \sum_{i,j,k,\ell} \left[(1-\alpha)M^r[i,j] + \alpha L^r[i,k,j,\ell]\right] \pi_{ij}\pi_{k\ell},$$

Suppose the cost inputs are perturbed as $M^r \mapsto \widetilde{M^r}$ and $L^r \mapsto \widetilde{L^r}$, and define

$$\Delta_M = \|M^r - \widetilde{M^r}\|_\infty, \quad \Delta_L = \|L^r - \widetilde{L^r}\|_\infty.$$

Let $\pi^*$ be an optimal coupling for $D_\alpha(M^r, L^r)$, and $\widetilde{\pi}^*$ be the optimal coupling for $D_\alpha(\widetilde{M^r}, \widetilde{L^r})$, then we observe that:

$$
\begin{aligned}
D_\alpha(M^r, L^r) - D_\alpha(\widetilde{M^r}, \widetilde{L^r}) &\le D_\alpha(M^r, L^r; \widetilde{\pi}^*) - D_\alpha(\widetilde{M^r}, \widetilde{L^r}; \widetilde{\pi}^*) \\
&= \sum_{i,j,k,\ell} \left[(1-\alpha)(M^r[i,j] - \widetilde{M^r}[i,j]) + \alpha(L^r[i,k,j,\ell] - \widetilde{L^r}[i,k,j,\ell])\right] \widetilde{\pi}_{ij}^* \widetilde{\pi}_{k\ell}^* \\
&\le \sum_{i,j,k,\ell} \left[(1-\alpha)\Delta_M + \alpha\Delta_L\right] \widetilde{\pi}_{ij}^* \widetilde{\pi}_{k\ell}^* \\
&= (1-\alpha)\Delta_M + \alpha\Delta_L,
\end{aligned}
$$

since $\sum_{i,j} \widetilde{\pi}_{ij}^* = \sum_{k,\ell} \widetilde{\pi}_{k\ell}^* = 1$.

By symmetry, the reverse inequality also holds. Hence,

$$\left| D_\alpha(M^r, L^r) - D_\alpha(\widetilde{M^r}, \widetilde{L^r}) \right| \le (1-\alpha)\Delta_M + \alpha\Delta_L.$$

## 1.2 PROOF OF THEOREM 4.1

**Theorem 4.1** *(Lipschitz Continuity of the Proposed Uncertainty Measure)*

Let $U(q)$ denote the uncertainty score for query $q$ with respect to a generated graph $G_a$ as the *anchor*, and $N-1$ independently generated graphs $\{G_r\}_{r \neq a}$ as *reference* graphs, defined as

$$U(q) = \frac{1}{N-1} \sum_{r \neq a} \mathrm{D}_\alpha(M^r, L^r),$$

where $(M^r, L^r)$ are the semantic cost matrices and structural cost tensors between $G_a$ and $G_r$. Then $U(q)$ is Lipschitz continuous with respect to the collection $\{M^r, L^r\}_{r \neq a}$.

**Proof.** Recall that each alignment distance $\mathrm{D}_\alpha(M^r, L^r)$ in $U(q)$ is Lipschitz continuous by **Lemma 4.1**:

$$\left| \mathrm{D}_\alpha(M^r, L^r) - \mathrm{D}_\alpha(\widetilde{M}^r, \widetilde{L}^r) \right| \leq (1-\alpha)\|M^r - \widetilde{M}^r\|_\infty + \alpha \|L^r - \widetilde{L}^r\|_\infty.$$

Now define, for each $r$,

$$\Delta_M^r = \|M^r - \widetilde{M}^r\|_\infty, \qquad \Delta_L^r = \|L^r - \widetilde{L}^r\|_\infty.$$

Then

$$\left| U(q) - \widetilde{U}(q) \right| = \frac{1}{N-1} \left| \sum_{r \neq a} \left[ \mathrm{D}_\alpha(M^r, L^r) - \mathrm{D}_\alpha(\widetilde{M}^r, \widetilde{L}^r) \right] \right|$$

$$\leq \frac{1}{N-1} \sum_{r \neq a} \left[ (1-\alpha)\,\Delta_M^r + \alpha\,\Delta_L^r \right]$$

$$\leq (1-\alpha) \max_{r \neq a} \Delta_M^r + \alpha \max_{r \neq a} \Delta_L^r.$$

Since this bound depends linearly on the perturbations $\{M^r, L^r\}_{r \neq a}$, it shows that $U(q)$ is Lipschitz continuous in the collection $\{M^r, L^r\}$, as claimed.

## 1.3 PROOF OF THEOREM 4.2

**Theorem 4.2** *(Convergence of the Uncertainty Measure)* Under general boundedness assumptions on the alignment distances, the uncertainty score $U(q)$ defined exponentially converges around the true mean $\mathrm{E}[U(q)]$, specifically:

$$\mathbb{P}\left[ |U(q) - \mathbb{E}[U(q)]| > \epsilon \right] \leq 2 \exp\left( -\frac{2(N-1)\epsilon^2}{D^2} \right),$$

for any $\epsilon$ and some constant $D > 0$ that depends on the graph generation inconsistency.

**Proof.** Let $G_a$ be the fixed anchor graph, and let $G_1, \ldots, G_{N-1}$ be the $N-1$ reference graphs, each sampled independently from the LLM's output distribution. Recall

$$U(q) = \frac{1}{N-1} \sum_{r=1}^{N-1} D_\alpha(M^r, L^r),$$

where $D_\alpha(M^r, L^r) = \mathrm{D}_\alpha(G_a, G_r)$ is the alignment distance between the anchor and the $r$-th reference. We further represent the uncertainty measure computed with graphs $\{G_1, \cdots, G_r, \cdots, G_{N-1}\}$ as $U(q; G_1, \ldots, G_r, \ldots, G_{N-1})$.

**Bounded differences.** Assume each term is bounded,

$$0 \leq D_\alpha(M^r, L^r) \leq D$$

for some constant $D > 0$. If we replace one reference graph $G_r$ by an independent draw $G_r'$, then only the $r$-th summand in $U(q)$ changes, and by boundedness,

$$\left| U(q; G_1, \ldots, G_r, \ldots, G_{N-1}) - U(q; G_1, \ldots, G_r', \ldots, G_{N-1}) \right| \leq \frac{D}{N-1}.$$

Hence $U(q)$ satisfies the bounded-differences property with constants $c_r = \frac{D}{N-1}$ for $r = 1, \ldots, N-1$.

**Application of McDiarmid's Inequality.** By McDiarmid's inequality for any $\epsilon > 0$,

$$\Pr\big[\, U(q) - \mathrm{E}[U(q)] \; > \; \epsilon \,\big] \;\leq\; \exp\Big(-\frac{2\epsilon^2}{\sum_{r=1}^{N-1} c_r^2}\Big) \;=\; \exp\Big(-\frac{2\,(N-1)\,\epsilon^2}{D^2}\Big).$$

A symmetric bound holds for $\Pr[\mathrm{E}[U(q)] - U(q) > \epsilon]$, so by the union bound,

$$\Pr\big[\big|U(q) - \mathrm{E}[U(q)]\big| > \epsilon\big] \;\leq\; 2\,\exp\Big(-\frac{2\,(N-1)\,\epsilon^2}{D^2}\Big),$$

as claimed.

***Remark 1:*** Importantly, $D$ is a property of the underlying generative process, specifically, the language model's consistency in producing structurally and semantically similar paragraphs in response to the same query. Lower values of $D$ indicate that the model tends to generate paragraphs that are more coherent and uniform in their graph representations, thereby enabling faster convergence and more stable uncertainty estimates.

## 2  GAUSS−ATOMIC : EXTENDING GAUSS TO ATOMIC-FACT UNCERTAINTY

---
**Algorithm 1** GAUSS-atomic

---
**Require:** LLM $\mathcal{M}$, query $q$, atomic extractor $\mathcal{M}_{\mathrm{atomic}}$, entailment model $\mathcal{M}_{\mathrm{entail}}$, sample count $N$
1: Sample $\{P_1, \ldots, P_N\} \sim \mathcal{M}(q)$
2: Anchor $P_a \leftarrow P_1$
3: Extract anchor facts $\{v_i\}_{i=1}^{n_a} \leftarrow \mathcal{M}_{\mathrm{atomic}}(P_a)$
4: **for** $r = 2, \ldots, N$ **do**
5:   Extract semantic graph of $P_r$ and compute

$$\big(M^r, \pi^r\big) \;\xleftarrow{\text{GAUSS}}\; \big(G_a, G_r\big),$$

6: **end for**
7: **for** $i = 1, \ldots, n_a$ **do**
8:

$$U_{\mathrm{fact}}(i) \leftarrow \tfrac{1}{N-1}\sum_{r=2}^{N} \cdot\frac{1}{n_r}\sum_{j=1}^{n_r}(1 - \pi_{ij}^r)\, M^r[i,j]$$

9: **end for**
10: Compute calibration metrics (AUROC, AUARC, etc) on $\{U_{\mathrm{fact}}(i)\}$

---

While GAUSS produces a single uncertainty score at the paragraph level, its design naturally lends itself to fine-grained extensions. We propose GAUSS-atomic, a fact-level variant of GAUSS, which "zooms in" on each atomic fact within the anchor paragraph and quantifies its uncertainty by evaluating alignment costs against reference generations. Specifically, GAUSS-atomic assigns an uncertainty score to each atomic fact by jointly considering its structural role and semantic correspondence within the broader context of the paragraph, thereby preserving the interpretability and rigor of the original GAUSS framework at a finer resolution.

Concretely, given a prompt $q$ and an LLM $\mathcal{M}$, we:

1. Draw $N$ independent long–form samples $\{P_1, \ldots, P_N\} \sim \mathcal{M}(q)$.

2. Designate $P_a = P_1$ as the *anchor*, and extract its atomic facts

$$\{f_i\}_{i=1}^{n_a} \;=\; \mathcal{M}_{\mathrm{atomic}}(P_a).$$

3. For each reference $P_r$ $(r = 2, \ldots, N)$:
   - Extract its semantic graph and compute the $\mathrm{D}_\alpha$ -alignment coupling $\pi^r \in \mathrm{R}^{n_a \times n_r}$ and cost matrix $M^r[i,j]$ between anchor nodes $v_i$ and reference nodes $w_j$.

4. Finally, for each anchor fact $v_i$ we define its atomic-fact uncertainty

$$U_{\mathrm{fact}}(i) \;=\; \frac{1}{N-1}\sum_{r=2}^{N} \cdot\frac{1}{n_r}\sum_{j=1}^{n_r}(1 - \pi_{ij}^r)\, M^r[i,j]$$

5. These $\{U_{\text{fact}}(i)\}$ can then be evaluated against ground-truth veracity labels to obtain calibration metrics like the AUROC, AUARC etc.

We provide the algorithmic flow for `GAUSS-atomic` in Algorithm 1. We illustrate the benefits of using `GAUSS` over other approaches in Table 1.

Table 1: Atomic calibration performance (AUROC ↑, AUARC ↑) on LongFact and WildHallu for four methods across five LLMs.

| Method / Model | LongFact | | WildHallu | |
|---|---|---|---|---|
| | AUROC | AUARC | AUROC | AUARC |
| **falcon-7b-instruct** | | | | |
| GEN-BINARY | 0.8462 | 0.9451 | 0.7859 | 0.8753 |
| DIS-RATING | 0.5143 | 0.8676 | 0.5032 | 0.7337 |
| DIS-SINGLE | 0.5000 | 0.8304 | 0.5001 | 0.7358 |
| GAUSS-atomic | **0.8600** | **0.9575** | 0.7853 | **0.8757** |
| **llama3-8b-instruct** | | | | |
| GEN-BINARY | 0.7205 | 0.9606 | 0.7876 | 0.9378 |
| DIS-RATING | 0.6839 | 0.9557 | 0.7479 | 0.9310 |
| DIS-SINGLE | 0.7560 | 0.9680 | **0.7984** | **0.9506** |
| GAUSS-atomic | **0.7616** | **0.9687** | 0.7847 | 0.9395 |
| **qwen2-7b-instruct** | | | | |
| GEN-BINARY | 0.7449 | 0.9587 | 0.7818 | 0.9298 |
| DIS-RATING | 0.6606 | 0.9404 | 0.6976 | 0.9025 |
| DIS-SINGLE | 0.7709 | 0.9657 | 0.7821 | 0.9334 |
| GAUSS-atomic | **0.7917** | **0.9674** | **0.7855** | **0.9382** |
| **qwen2-57b-instruct** | | | | |
| GEN-BINARY | 0.7192 | 0.9696 | 0.7762 | 0.9415 |
| DIS-RATING | 0.7530 | 0.9689 | 0.7494 | 0.9352 |
| DIS-SINGLE | 0.7830 | 0.9781 | 0.7685 | 0.9421 |
| GAUSS-atomic | **0.7920** | **0.9776** | **0.7691** | **0.9424** |
| **mistral-7b-instruct** | | | | |
| GEN-BINARY | 0.7372 | 0.9659 | 0.7556 | 0.9282 |
| DIS-RATING | 0.6821 | 0.9551 | 0.6674 | 0.8940 |
| DIS-SINGLE | 0.5472 | 0.9163 | 0.6065 | 0.8539 |
| GAUSS-atomic | **0.7612** | **0.9688** | **0.7689** | **0.9301** |

## 3 FILTERING ATOMIC FACTS VIA `GAUSS-ATOMIC`

One application of the `GAUSS-atomic` uncertainty scores is to improve the overall reliability of the generated content by *filtering out* the most uncertain atomic facts. Concretely, given a generated paragraph $P$, we:

1. Decompose $P$ into $n_a$ atomic facts $\{f_i\}_{i=1}^{n_a}$.

2. For each fact $f_i$:
   - Compute its uncertainty score $u_i = U_{\text{fact}}(i)$ via `GAUSS-atomic`.
   - Obtain its binary veracity label $s_i \in \{0, 1\}$ via the SAFE fact-checking module.

3. Choose a filtering level $k\%$: discard the top $k\%$ most-uncertain facts.

4. Let $\tau = \text{percentile}(\{u_i\}, 100 - k)$. Keep only $\mathcal{S} = \{\, i : u_i \leq \tau \,\}$, and compute the *filtered mean veracity*

$$\bar{v}_{\text{filtered}} = \frac{1}{|\mathcal{S}|} \sum_{i \in \mathcal{S}} v_i.$$

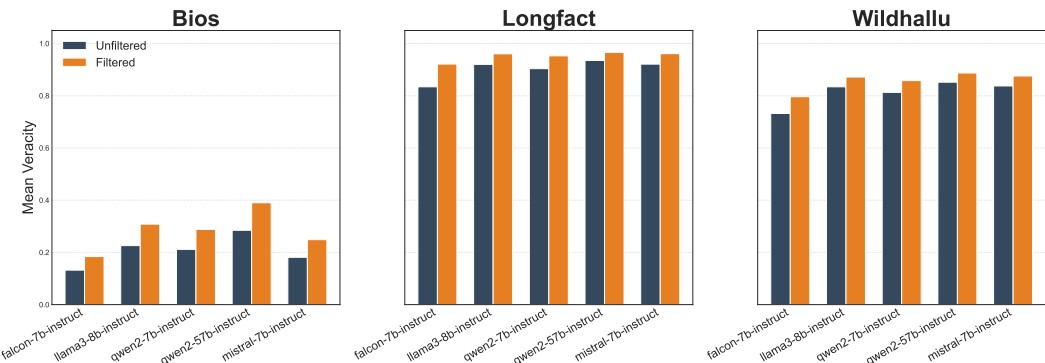

Figure 1: The increase in mean veracity $\bar{v}_{\text{filtered}}$ of the uncertainty based filtering - over the unfiltered-case $\bar{v}_{\text{all}}$

5. Compare $\bar{v}_{\text{filtered}}$ against the unfiltered mean $\bar{v}_{\text{all}} = \frac{1}{n_a} \sum_i s_i$ to measure the gain in factuality by uncertainty based filtering.

---

**Algorithm 2** Atomic-Fact Filtering via `GAUSS-atomic`

---

**Require:** Paragraph $P$, filtering rate $k \in [0, 100]$
1: $\{f_i\} \leftarrow$ `decompose_into_atomic_facts`$(P)$
2: $n_a \leftarrow |\{f_i\}|$
3: **for** $i = 1$ to $n_a$ **do**
4:    $u_i \leftarrow$ `GAUSS-atomic` $(f_i)$
5:    $s_i \leftarrow$ `SAFE_veracity`$(f_i)$
6: **end for**
7: $\bar{v}_{\text{all}} \leftarrow \frac{1}{n_a} \sum_{i=1}^{n_a} s_i$
8: $\tau \leftarrow \text{percentile}(\{u_i\}, 100 - k)$
9: $\mathcal{S} \leftarrow \{ i : u_i \leq \tau \}$
10: $\bar{v}_{\text{filtered}} \leftarrow \frac{1}{|\mathcal{S}|} \sum_{i \in \mathcal{S}} v_i$
11:
12: **return** $(\bar{v}_{\text{all}}, \bar{v}_{\text{filtered}})$

---

This method provides a simple, yet practical guide to obtain filtered atomic facts that have a strong guarantee to have higher veracity. We provide the algorithmic flow of the uncertainty based filtering approach in Algorithm 2 and the results of this approach in Figure 1.

## 4   ALTERNATIVE NOTIONS OF STRUCTURAL COST

Beyond simple pairwise local distances, we can capture richer, global connectivity in our semantic graphs by using the *heat kernel* of the graph Laplacian. Concretely, given a semantic graph

$$G_i = (V_i, E_i, C_i, \ell_f)$$

with $n_i$ nodes, let $L_i \in \mathbb{R}^{n_i \times n_i}$ be its (normalized) graph Laplacian. The heat kernel at time $\tau > 0$ is defined as

$$H_i(\tau) = \exp(-\tau L_i) \in \mathbb{R}^{n_i \times n_i}, \tag{4.1}$$

where $\exp$ denotes the matrix exponential. Entries $H_i(\tau)_{k,\ell}$ encode the flux from node $k$ to node $\ell$ over time $\tau$, thereby reflecting both local and multi-hop structural relationships.

To incorporate this into our alignment distance, we replace the original structural cost tensor $L^r$ (based on raw distance or cosine-dissimilarity) by a *heat-kernel cost tensor* $\widehat{L}^r(\tau)$ defined as

$$\widehat{L}^r(\tau)[i, j, k, \ell] = |H_a(\tau)_{i,k} - H_r(\tau)_{j,\ell}|. \tag{4.2}$$

Here $H_a(\tau)$ and $H_r(\tau)$ are the heat kernels of the anchor and reference graphs, respectively. We utilize $\tau = 0.99$ for the results shown in Table 2.

Finally, the full alignment distance between $G_a$ and $G_r$ becomes

$$\mathrm{D}_\alpha^{\mathrm{heat}}\big(G_a, G_r; \tau\big) = \min_{\pi \in \Pi} \sum_{i,j,k,\ell} \Big[(1 - \alpha)\, M_{i,j}^r \;+\; \alpha\, \widehat{L}_{i,j,k,\ell}^r(\tau)\Big]\, \pi_{ij}\, \pi_{k\ell}, \qquad (4.3)$$

which we then plug into the uncertainty measure $U(q)$.

Table 2: Comparison of GAUSS-kernel ($\tau = 0.99$) and GAUSS on atomic-calibration metrics (SC, PC, UCCE, QCCE) for LongFact and WildHallu (lower is better).

| Model / Method | LongFact | | | | WildHallu | | | |
|---|---|---|---|---|---|---|---|---|
| | SC | PC | UCCE | QCCE | SC | PC | UCCE | QCCE |
| **falcon-7b-instruct** | | | | | | | | |
| GAUSS-kernel | -0.6428 | -0.6767 | 0.2089 | 0.2777 | -0.7624 | -0.7682 | 0.1809 | 0.2078 |
| GAUSS | -0.6555 | -0.6915 | 0.1817 | 0.2199 | -0.7565 | -0.7616 | 0.1470 | 0.1978 |
| **llama3-8b-instruct** | | | | | | | | |
| GAUSS-kernel | -0.4038 | -0.4224 | 0.1702 | 0.2515 | -0.6789 | -0.7080 | 0.2133 | 0.2439 |
| GAUSS | -0.4433 | -0.4505 | 0.1613 | 0.2535 | -0.6808 | -0.7144 | 0.2048 | 0.2624 |
| **qwen2-7b-instruct** | | | | | | | | |
| GAUSS-kernel | -0.4891 | -0.5155 | 0.1522 | 0.2869 | -0.6999 | -0.7017 | 0.1799 | 0.2378 |
| GAUSS | -0.4979 | -0.5233 | 0.1403 | 0.2362 | -0.7072 | -0.7154 | 0.1798 | 0.2212 |
| **qwen2-57b-instruct** | | | | | | | | |
| GAUSS-kernel | -0.3582 | -0.4006 | 0.2367 | 0.3502 | -0.6644 | -0.7190 | 0.1966 | 0.2664 |
| GAUSS | -0.4226 | -0.4615 | 0.1844 | 0.3111 | -0.6852 | -0.7032 | 0.2061 | 0.2043 |
| **mistral-7b-instruct** | | | | | | | | |
| GAUSS-kernel | -0.4064 | -0.4491 | 0.2151 | 0.2815 | -0.6940 | -0.7591 | 0.1867 | 0.2243 |
| GAUSS | -0.4408 | -0.4678 | 0.2525 | 0.2672 | -0.6949 | -0.7584 | 0.1784 | 0.2310 |

We observe from Table 2 that employing straightforward semantic distance representations, such as adjacency matrices, to construct the structural cost tensor in GAUSS yields more effective calibration performance.

## 5 ROBUSTNESS TO SEMANTIC EMBEDDING VARIATIONS

Table 3: Atomic-calibration metrics (SC, PC, UCCE, QCCE) for two embedding variants of GAUSS.

| Model / Embedding | Bios | | | | LongFact | | | | WildHallu | | | |
|---|---|---|---|---|---|---|---|---|---|---|---|---|
| | SC | PC | UCCE | QCCE | SC | PC | UCCE | QCCE | SC | PC | UCCE | QCCE |
| **falcon-7b-instruct** | | | | | | | | | | | | |
| GAUSS(all-mpnet-base-v2) | -0.4118 | -0.3321 | 0.1680 | 0.1845 | -0.6555 | -0.6915 | 0.1817 | 0.2199 | -0.7565 | -0.7616 | 0.1470 | 0.1978 |
| GAUSS(stsb-roberta-base) | -0.4080 | -0.3321 | 0.1832 | 0.1659 | -0.6727 | -0.7045 | 0.1974 | 0.2501 | -0.7517 | -0.7570 | 0.1829 | 0.2142 |
| **llama3-8b-instruct** | | | | | | | | | | | | |
| GAUSS(all-mpnet-base-v2) | -0.7066 | -0.7080 | 0.1035 | 0.1426 | -0.4433 | -0.4505 | 0.1613 | 0.2535 | -0.6808 | -0.7144 | 0.2048 | 0.2624 |
| GAUSS(stsb-roberta-base) | -0.7070 | -0.7045 | 0.1348 | 0.1035 | -0.5058 | -0.5240 | 0.1753 | 0.2559 | -0.6894 | -0.7141 | 0.2217 | 0.2707 |
| **qwen2-7b-instruct** | | | | | | | | | | | | |
| GAUSS(all-mpnet-base-v2) | -0.6915 | -0.7114 | 0.1606 | 0.1505 | -0.4979 | -0.5233 | 0.1403 | 0.2362 | -0.7072 | -0.7154 | 0.1798 | 0.2212 |
| GAUSS(stsb-roberta-base) | -0.6953 | -0.7143 | 0.1420 | 0.1664 | -0.5618 | -0.5892 | 0.1623 | 0.2617 | -0.7108 | -0.7178 | 0.1808 | 0.2385 |
| **qwen2-57b-instruct** | | | | | | | | | | | | |
| GAUSS(all-mpnet-base-v2) | -0.6991 | -0.7018 | 0.1328 | 0.1092 | -0.4226 | -0.4615 | 0.1844 | 0.3111 | -0.6852 | -0.7032 | 0.2061 | 0.2043 |
| GAUSS(stsb-roberta-base) | -0.6928 | -0.6995 | 0.0746 | 0.0952 | -0.4636 | -0.5311 | 0.2055 | 0.3329 | -0.6658 | -0.7186 | 0.2038 | 0.2603 |
| **mistral-7b-instruct** | | | | | | | | | | | | |
| GAUSS(all-mpnet-base-v2) | -0.6643 | -0.6766 | 0.1443 | 0.1407 | -0.4408 | -0.4678 | 0.2525 | 0.2672 | -0.6949 | -0.7584 | 0.1784 | 0.2310 |
| GAUSS(stsb-roberta-base) | -0.6648 | -0.6663 | 0.1261 | 0.1453 | -0.4820 | -0.4754 | 0.2001 | 0.2819 | -0.6963 | -0.7596 | 0.1716 | 0.2250 |

In order to explore the robustness of `GAUSS` to perturbations in the embeddings, we experiment with two different embedding models with the same embedding size. These models are `all-mpnet-base-v2` and `stsb-roberta-base`.

**all-mpnet-base-v2**   This model is fine-tuned on a broad mixture of semantic similarity datasets (e.g., STS, QuoraQP), and generates 768-dimensional embeddings. It is widely regarded as one of the strongest general-purpose sentence encoders in terms of semantic textual similarity (STS) performance.

**stsb-roberta-base**   This model builds on the RoBERTa-base encoder, fine-tuned on the STS Benchmark using a Siamese-BERT (SBERT) architecture. It also produces 768-dimensional embeddings that are optimized for capturing fine-grained relational semantics between sentence pairs.

Despite architectural and training differences, Table 3 reveals that both models yield comparable performance in calibration metrics across datasets. This empirical stability highlights a key theoretical property of our framework: as formalized in **Theorem 4.1**, the proposed uncertainty measure is Lipschitz continuous in its semantic cost inputs. Hence, bounded perturbations in node embeddings (e.g., due to switching between embedding models) induce only minor, controlled changes in uncertainty scores and thereby small changes in calibration. This result not only ensures robustness to sentence encoder variations but also validates the practical reliability of `GAUSS` across diverse embedding backbones.

# 6   ADDITIONAL EVALUATION DATASETS: ELI5 AND SCIQA

Table 4: Calibration metrics (SC, PC, UCCE, QCCE) on ELI5 and SciQA.

| Model / Method | ELI5 | | | | SciQA | | | |
|---|---|---|---|---|---|---|---|---|
| | SC | PC | UCCE | QCCE | SC | PC | UCCE | QCCE |
| **falcon-7b-instruct** | | | | | | | | |
| DIS-RATING | 0.1079 | 0.1261 | **0.0373** | 0.2185 | 0.0621 | 0.0118 | 0.0567 | 0.1140 |
| DIS-SINGLE | 0.0283 | -0.0335 | 0.1533 | 0.4962 | 0.0430 | 0.0707 | **0.0000** | 0.5490 |
| GEN-BINARY | -0.5586 | -0.5696 | 0.1470 | 0.1929 | -0.5054 | -0.5465 | 0.0828 | 0.1124 |
| LUQ | -0.3164 | -0.3051 | 0.1219 | **0.1622** | -0.1878 | -0.2671 | 0.0924 | **0.0950** |
| GAUSS | **-0.6022** | **-0.5704** | 0.1237 | 0.1671 | **-0.5503** | **-0.6021** | 0.0744 | 0.1568 |
| **mistral-7b-instruct** | | | | | | | | |
| DIS-RATING | -0.3568 | -0.3185 | **0.1195** | **0.1577** | -0.1706 | -0.1657 | 0.1318 | 0.1854 |
| DIS-SINGLE | 0.1122 | 0.1045 | 0.1402 | 0.1927 | -0.0096 | -0.0969 | 0.1537 | 0.1976 |
| GEN-BINARY | -0.3662 | -0.3186 | 0.1480 | 0.1653 | -0.3002 | **-0.4275** | 0.1832 | **0.0976** |
| LUQ | -0.1271 | -0.0755 | 0.1761 | 0.1880 | -0.1071 | 0.0317 | 0.1372 | 0.1659 |
| GAUSS | **-0.3775** | **-0.3214** | 0.1699 | 0.2242 | **-0.3102** | -0.4220 | **0.1177** | 0.0995 |

To further assess our uncertainty quantification framework, we experiment on two publicly available long-form QA benchmarks:

**ELI5**   The ELI5 dataset [1] is drawn from the "Explain Like I'm Five" subreddit. It covers a broad range of user-curated explanatory queries, making it a challenging testbed for paragraph-level UQ.

**SciQA**   SciQA [4] is a science-focused QA collection of questions sourced from elementary and middle-school science curricula. Its domain specificity and factual rigor stress-test our uncertainty estimates in technical contexts.

We observe from Table 4 that `GAUSS` produces consistently better correlation with the factuality values.

# 7 CONVERGENCE OF $U(q)$ WITH SAMPLE SIZE AND LLM CONSISTENCY

We empirically examine the convergence behavior of the uncertainty measure $U(q)$ as a function of the number of reference samples $N$. According to Theorem 4.2, $U(q)$ concentrates around its expected value $\mathbb{E}[U(q)]$ at an exponential rate, with respect to the number of reference paragraphs $(N-1)$ and the graph generation inconsistency factor $D$ of the underlying LLM. As shown in Figure 2, the deviation $|U(q) - \mathbb{E}[U(q)]|$ diminishes as $N$ increases. Convergence is also faster for some models over others.

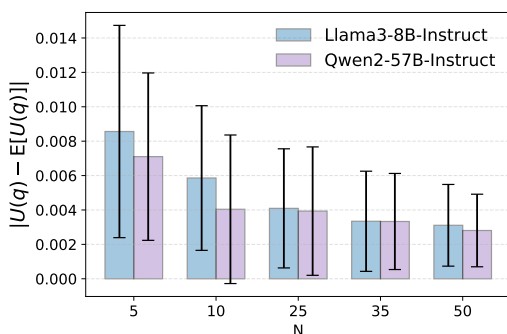

Figure 2: Convergence of the uncertainty measure $U(q)$ to its expected value as the number of reference samples $N$ increases.

To further quantify this behavior, we interpret the parameter $D$ as an intrinsic property of an LLM that governs the rate at which its empirical uncertainty estimates converge to the true expected uncertainty. For a fixed failure probability i.e. $\mathbb{P}[|U(q) - \mathbb{E}[U(q)]| > \epsilon] = \delta$, sample size $N$, and tolerance $\epsilon$, the high-probability upper bound on $D$ is given by:

$$D = \epsilon \sqrt{\frac{2(N-1)}{\ln(2/\delta)}}.$$

By setting $\delta = 0.05$ and averaging over multiple runs, we compute the mean upper bound of $D$ for two representative models:

$$D_{\texttt{Qwen-2-57B-Instruct}} \approx 0.0129, \quad D_{\texttt{LLaMA3-8B-Instruct}} \approx 0.0149.$$

Lower values of $D$ indicate LLMs with reduced graph inconsistency and faster convergence of their uncertainty estimates. This formulation offers a model-agnostic metric to compare LLMs based on their consistency in generating structurally and semantically coherent outputs across sampled generations.

# 8 CHOICE OF ANCHOR PARAGRAPH

Our uncertainty estimation procedure requires selecting an anchor paragraph $G_a$ to compare against the remaining generations for a given query. By default, we follow prior work [5, 6], which uses the first generated paragraph as the anchor. While simple and reproducible, this choice may introduce variability in the resulting uncertainty scores. We thereby perform experiments with different anchor paragraphs below.

EMPIRICAL STABILITY ON BIOS / LLAMA3-8B-INSTRUCT

To evaluate the robustness of our setup to anchor choice, we conduct two empirical analyses on the Bios dataset.

**1. Coefficient of Variation.** For each query, we compute the uncertainty score $U(q)$ using every paragraph in the set as the anchor, and record the coefficient of variation:

$$\text{CV} = \frac{\sigma(U(q))}{\mu(U(q))}.$$

where $\mu(\cdot), \sigma(\cdot)$ are the empirical mean and standard deviation across different anchor paragraphs for a given query. Averaged across queries, we find $\overline{\text{CV}} = 0.2333$, suggesting relatively low spread in uncertainty estimates.

**2. Correlation-Coefficient Variability.** We also assess how anchor choice affects agreement with ground-truth veracity labels by computing the variability in correlation metrics across anchors:

| Metric | Mean | Std Dev |
|--------|------|---------|
| SC | $-0.6844$ | 0.3799 |
| PC | $-0.6826$ | 0.4077 |

These results show mild sensitivity to anchor selection in current settings. Nevertheless, we recognize that more principled strategies may further enhance stability.

One such alternative involves choosing the paragraph whose semantic graph is, on average, closest to all others, thereby minimizing structural deviation and reducing outlier influence:

$$G_a = \arg \min_{G \in \mathcal{G}_i} \sum_r D_\alpha(G, G_r).$$

Another lightweight strategy would be to select the paragraph with the median number of atomic fact nodes, which often approximates the central structure without requiring pairwise graph comparisons. We leave investigation into these and other anchor-selection strategies to future work.

# 9 COMPUTATIONAL ANALYSIS OF GAUSS

We analyze the computational cost and practical runtime of GAUSS. In practice, the runtime of GAUSS is dominated by three components: graph construction, entailment computation, and alignment.

Graph construction is efficient and uses lightweight embedding models such as `all-mpnet-base-v2`, which have modest parameter sizes and fast inference.

GAUSS performs graph alignment between the anchor paragraph and each of the $N - 1$ sampled reference paragraphs which involves the computation of the structural cost tensor. Overall the convex optimization process to find the final alignment distance between the anchor and all references has a complexity of

$$\mathcal{O}\left(N \cdot n_a^2 \cdot n_r^2\right),$$

where $N - 1$ is the number of references, $n_a$ is the number of atomic facts in the anchor, and $n_r$ is the number in each reference graph. Alignment is fully parallelizable and offloaded to an an AMD EPYC 7413 CPU.

Entailment, which is common to GAUSS and all other baselines, is performed using larger models like Qwen2-32B-Instruct and constitutes the primary computational bottleneck.

We compare GAUSS with Gen-Binary, which also depends on entailment checks, under a consistent hardware setup (Nvidia A100 GPUs for embedding and entailment). The runtimes across datasets are reported below:

| Dataset | # Queries | GAUSS Runtime (mins) | Gen-Binary Runtime (mins) |
|---------|-----------|----------------------|---------------------------|
| WildHallu | 500 | $\approx 35$ | $\approx 30$ |
| LongFact | 500 | $\approx 35$ | $\approx 30$ |
| Bios | 183 | $\approx 10$ | $\approx 8$ |

Since the dominant cost for both methods lies in entailment inference, their runtimes are comparable. However, GAUSS yields stronger alignment with ground-truth veracity, achieving higher Spearman and Pearson correlations, making it more effective without added computational overhead.

## 10 EXTENDING GAUSS WITH CAUSAL DEPENDENCY MODELING

To extend GAUSS to paragraphs with logical flow, we integrate causal modeling into its structural alignment. Specifically, we construct a causal structure matrix $A \in \{0,1\}^{n \times n}$ over atomic facts, where:

$$A(i,k) = \begin{cases} 1 & \text{if } v_i \text{ causally precedes or leads to } v_k, \\ 0 & \text{otherwise.} \end{cases}$$

This yields a directed, non-symmetric adjacency matrix inferred via a separate language model. We define a fused structural cost tensor that incorporates both semantic and causal signals:

$$C_a(i,k) = \left[1 - \cos\left(l_f(v_i),\, l_f(v_k)\right)\right] + \left[1 - A(i,k)\right],$$

where $l_f(v)$ denotes the embedding of atomic fact $v$. The first term penalizes semantic dissimilarity, and the second penalizes missing or reversed causal direction. We embed this cost tensor into Equation 4.3 in the manuscript and compute the uncertainty measure. The resulting uncertainty reflects both semantic cohesion and causal consistency across the paragraph.

To calibrate the uncertainty measure, we use a domain-specific anchor paragraph and perform the following:

- A language model rates the anchor paragraph's structural alignment with a golden reference on a scale $s_{\text{struct}} \in [0,1]$.

- The mean factuality score across atomic facts in the anchor paragraph is calculated $s_{\text{fact}} \in [0,1]$.

- The overall anchor score is: $s_{\text{anchor}} = \frac{1}{2}(s_{\text{struct}} + s_{\text{fact}})$.

We calibrate the uncertainty measure with $s_{\text{anchor}}$ to obtain SC and PC values which are reported below. We perform these experiments on a small custom dataset requiring causal paragraphs. We expect that an LLM that produces structurally, logically and semantically similar paragraphs should have a high $s_{\text{anchor}}$ for the anchor paragraph.

Table 5: Correlation between $s_{\text{anchor}}$ and uncertainty measure of GAUSS.

| Model | SC | PC |
|---|---|---|
| Qwen2-7B-Instruct | $-0.821$ | $-0.987$ |
| Llama-3-8B-Instruct | $-0.900$ | $-0.867$ |

## 11 ADDITIONAL EXPERIMENTAL DETAILS

All experiments were conducted using NVIDIA A100 GPUs. For generating responses from large language models (LLMs), we employed a sampling strategy with a temperature of 1.0, top-p of 0.95, and a maximum token limit of 512. Each model was prompted to produce 20 distinct responses per input, aligning with the evaluation protocols for GAUSS LUQ, and GEN-BINARY. In computing the semantic cost matrix $M^r$, we binarize the semantic cost matrix by setting entries with values $\geq 0.6$ to 1 and the rest to 0, thereby emphasizing stronger semantic alignments. For the SAFE framework, relevant web pages were retrieved using the Serper API, a high-performance Google Search API known for delivering real-time search results with unparalleled speed.

## 12 PROMPT DESIGN

We illustrate the prompt design for $\mathcal{M}_{\text{atomic}}$ in Table 6.

We also illustrate the prompting approach for the $\mathcal{M}_{\text{entail}}$ in Table 7.

Below is the prompt for the SAFE framework in Table 8.

Table 6: Prompt template for the $\mathcal{M}_{\text{atomic}}$ fact-decomposition model.

---

**$\mathcal{M}_{\text{atomic}}$ Prompt**

Please break down the following passage into independent fact pieces.

**Step 1:** For each sentence, split it into atomic facts, each containing exactly one subject–verb–object triple. If no explicit verb appears, use "be" as the predicate.

**Step 2:** Output every fact piece on its own line, prefixed with ### and without any extra formatting.

**Step 3:** Ensure each fact is fully self-contained: avoid pronouns (he, she, it, this, that), and always repeat the original noun.

**Examples:**
Michael Collins (born October 31, 1930) is a retired American astronaut and test pilot...
```
### Michael Collins was born on October 31, 1930.
### Michael Collins is retired.
```
...
• *League of Legends (often abbreviated as LoL) is a multiplayer online battle arena video game...*
```
### League of Legends is a video game.
```
...
• *Emory University has a strong athletics program, competing in the NCAA Division I ACC...*
```
### Emory University has a strong athletics program.
```
...

Now it's your turn. Here is the passage:
$$\{\texttt{passage\_text}\}$$

Return only the list of prefixed fact pieces.

---

Table 7: Prompt template for the $\mathcal{M}_{\text{entail}}$ support-checking model.

---

**$\mathcal{M}_{\text{entail}}$ Prompt**

Paragraph:
{paragraph}

Atomic Fact:
{fact}

Is the above atomic fact *supported* by the given paragraph?

Answer solely from the context—do *not* rely on external knowledge.
Do *not* provide explanations.

**Output:** Either yes or no.
**Answer:**

---

Table 8: Prompt template for the SAFE fact-checking model.

---

**SAFE Fact-Checking Prompt**

Your task is to fact-check the following statement.
This statement is extracted from a passage about a specific subject (e.g., a person, place, or event).

Assign a veracity label:
- 'S' if the statement is factually correct.
- 'NS' if the statement is factually incorrect.

For example, given that we have the statement and evidence as such, output the veracity label output and your brief analysis as such:
**Statement**: Lebron James is a basketball player.
**Evidence**: Lebron James is an American professional basketball player for the Los Angeles Lakers of the NBA.
**Analysis**: Lebron James is an American professional basketball player, so this is correct.
**Output**: S
Pay close attention to numbers, dates, and other details.

Now for the statement and evidence below, output your brief analysis and veracity label output in the above described format:

- **Statement:** {atomic_fact}
- **Evidence:** {retrieved_evidence}
- **Output:** ¡your output here¿

---

## 13 LIMITATIONS AND FUTURE WORK

While GAUSS provides a principled and interpretable framework for uncertainty quantification in long-form LLM generation, there remain several areas that invite further investigation:

- **Computational cost of graph alignment.** The graph alignment distance involves computing the structural costs tensor with a computational cost of $O(n_a^2\, n_r^2)$ tensor, where $n_a$ and $n_r$ are the number of atomic facts in the anchor and reference paragraphs, respectively. This computation can be massively parallelized for fast computation for paragraphs with typical lengths.

- **Unified treatment of uncertainty.** At present, GAUSS captures overall variability in generated outputs without explicitly distinguishing between *epistemic* uncertainty (stemming from model limitations) and *aleatoric* uncertainty (reflecting inherent ambiguity in the input or data). Future extensions could incorporate mechanisms to disentangle and analyze these complementary aspects, providing more granular uncertainty quantification.

- **Simplified graph structure.** Our semantic graphs are currently constructed using undirected edges based on symmetric semantic similarity. While this captures structural coherence well, it does not yet encode directional or causal dependencies between facts such as temporal or inferential relationships. Enriching the graph representation with directed edges or causal signals may further improve uncertainty quantification.

## 14 BRIER SCORE AND EXPECTED CALIBRATION ERROR

We also report standard Brier score and expected calibration error (ECE) for the binarized veracity labels and the uncertainty score (which are typically defined for binary labels and probabilistic predictions). Our ground-truth veracity, however, is originally a continuous score $v(q) \in [0, 1]$ (fraction of supported atomic facts in the paragraph). To make it compatible with these binary calibration metrics, we first construct a binarized label

$$y(q) = \mathbf{1}\{v(q) > 0.5\},$$

treating paragraphs with factuality above $0.5$ as "high-factual" and the rest as "low-factual."

For each method $m$ and query $q$, we obtain a scalar uncertainty score $u_m(q)$ and interpret it as a probability after monotone rescaling to $[0, 1]$, with higher values corresponding to a higher probability of being incorrect (equivalently, lower probability of being correct). We then compute the *Brier score* for method $m$ in the standard way as the mean squared error between predicted probabilities and

binary labels, and the *expected calibration error* (ECE) using 10 equal-width bins over $[0, 1]$: in each bin we compare the average predicted probability with the empirical error rate and aggregate the absolute differences weighted by the bin frequencies. The resulting Brier and ECE values for GAUSS and all baselines are reported in Table 9.

| Model | Dataset | Brier ↓ | | ECE ↓ | |
|---|---|---|---|---|---|
| | | GAUSS | Gen-Binary | GAUSS | Gen-Binary |
| Falcon-7B-Instruct | Bios | 0.2110 | 0.4544 | 0.3856 | 0.6503 |
| | LongFact | 0.0654 | 0.0870 | 0.1164 | 0.1747 |
| | WildHallu | 0.1193 | 0.1199 | 0.0981 | 0.1297 |
| LLaMA3-8B-Instruct | Bios | 0.2006 | 0.4243 | 0.3278 | 0.5787 |
| | LongFact | 0.0112 | 0.0309 | 0.0944 | 0.1497 |
| | WildHallu | 0.0603 | 0.0976 | 0.0797 | 0.1693 |
| Qwen2-57B-Instruct | Bios | 0.2339 | 0.6868 | 0.3442 | 0.7349 |
| | LongFact | 0.0147 | 0.0274 | 0.0912 | 0.1137 |
| | WildHallu | 0.0645 | 0.0838 | 0.0425 | 0.1438 |

Table 9: Brier score and Expected Calibration Error (ECE) for GAUSS and Gen-Binary across models and datasets with binarized veracity labels. Lower is better.

## 15 PRACTICAL RUNTIME OF THE GAUSS DISTANCE METRIC

Recall that the GAUSS uncertainty score for a query $q$ is defined as

$$U(q) = \frac{1}{N-1} \sum_{r \neq a} D_\alpha(G_a, G_r),$$

where $G_a$ is the anchor graph and $\{G_r\}_{r \neq a}$ are the graphs induced by the remaining $N-1$ generations. Each term $D_\alpha(G_a, G_r)$ in this summation is independent and can therefore be computed fully in parallel across references, so the overall uncertainty computation scales well with the number of sampled paragraphs. Section 9 of the appendix reports end-to-end runtimes for the full GAUSS pipeline, from atomic fact decomposition to final uncertainty scores.

The fused Gromov–Wasserstein distance $D_\alpha(G_a, G_r)$ itself has a nominal complexity of $O(n_a^2 n_r^2)$ for graphs with $n_a$ and $n_r$ nodes, but in our setting $n_a, n_r \approx 50$ and the computation is fast in practice. This is largely because the underlying components of the cost, including the semantic term $M^r[i,j]$ and the structural term $C[i,k,j,\ell]$, are precomputed once per paragraph and reused across alignments. We report practical runtimes for computing $D_\alpha(G_a, G_r)$ in Table 10 in the appendix.

## 16 ATOMIC FACT EXTRACTION: NOISE ANALYSIS

**Robustness to the choice of atomic fact extractor.** In most long-form uncertainty methods, depending on an auxiliary LLM to decompose paragraphs into atomic facts, there lies a concern on the atomic fact extraction noise. In our framework, we denote this decomposition model by $M_{\text{atomic}}$ (Qwen2-32B-Instruct in the main experiments) and the corresponding prompt by $p$ (given in Table 6 of the Appendix). For a given response paragraph, the atomic fact list produced by $(M_{\text{atomic}}, p)$ is denoted by $AL_{\text{orig}}$. We emphasize that this decomposition step is shared across GAUSS and fact-level baselines such as LUQ, Gen-Binary, and Centrality.

To analyse the extraction noise, we perform an ablation study that probes the sensitivity of GAUSS to variations in the atomic fact extraction process along two axes: (i) *changing the decomposition model* $M_{\text{atomic}}$ while keeping the prompt $p$ fixed, and (ii) *changing the prompt* while keeping $M_{\text{atomic}}$ fixed. Concretely, for (i) we vary the model size (Qwen2-7B, Qwen2-57B) under the same prompt $p$, and for (ii) we fix $M_{\text{atomic}}$ and consider two prompt variants: removing some facts in the few-shot examples from Table 6, and removing all few-shot examples entirely.

Given an alternative atomic fact list AL for the same paragraph, we quantify its similarity to $AL_{\text{orig}}$ using two complementary metrics:

| Operation | Setting | Mean Runtime |
|---|---|---|
| FGW distance $D_\alpha(G_a, G_r)$ | (per pair, averaged over all datasets) | 29.20 ms |
| Uncertainty $U(q)$ (GAUSS) | Bios, $N = 20$ generations | 22.544 s |
| Uncertainty $U(q)$ (GAUSS) | LongFact, $N = 20$ generations | 30.207 s |
| Uncertainty $U(q)$ (GAUSS) | WildHallu, $N = 20$ generations | 32.392 s |

Table 10: Practical runtimes for the fused Gromov–Wasserstein distance $D_\alpha(G_a, G_r)$ and the GAUSS uncertainty computation $U(q)$ with $N = 20$ generations.

1. **LLM-based overlap score.** For each atomic fact $f \in$ AL, we query an LLM to decide whether $f$ is semantically present in $AL_{orig}$ (outputting 1 if supported and 0 otherwise). The overlap score between AL and $AL_{orig}$ is the mean of these binary labels over all $f \in$ AL. We report the average overlap score across all responses in a dataset; higher values indicate that the two decompositions capture essentially the same factual content.

2. **GAUSS graph distance.** Each atomic fact list (both AL and $AL_{orig}$) induces a semantic graph under the GAUSS construction. We compute the GAUSS alignment distance $D(AL, AL_{orig})$ between the two induced graphs and report the mean distance across all responses; lower values indicate more similar graph structure and hence a more stable decomposition.

Empirically, we find that *changing the prompt* (while holding $M_{atomic}$ fixed) leads to noticeably *lower* overlap scores and *higher* graph distances, reflecting the fact that very different prompting can change how aggressively the model splits or merges facts. In contrast, when we *change the model size* (Qwen2-7B / 32B / 57B) under a fixed, well-specified prompt $p$, the overlap scores remain high and the GAUSS graph distances remain small, indicating that the resulting atomic fact lists are very close both semantically and structurally.

| Setting | $M_{atomic}$ | Prompt | Overlap ↑ | Mean Graph Dist. ↓ |
|---|---|---|---|---|
| Varying model, fixed $p$ | Qwen2-7B-Instruct | $p$ (orig.) | 0.9619 | 0.1288 |
| | Qwen2.5-57B-Instruct | $p$ (orig.) | 0.9823 | 0.0412 |
| Varying prompt, fixed Qwen2.5-32B | Qwen2.5-32B-Instruct | $p$ (few-shot pruned) | 0.9591 | 0.1786 |
| | Qwen2.5-32B-Instruct | $p$ (no few-shot) | 0.9037 | 0.1960 |

Table 11: Overlap score and GAUSS graph distance between alternative atomic fact extraction configurations and the reference decomposition ($M_{atomic}$ = Qwen2.5-32B-Instruct, prompt $p$). Higher overlap and lower distance indicate closer agreement with the reference atomic fact list.

These results suggest that, in practice, once a stable prompt $p$ is chosen (such as the one we provide in Table 6), the choice of $M_{atomic}$ has only a modest effect on the induced semantic graphs and the resulting GAUSS uncertainty scores. While GAUSS, like all fact-level methods, necessarily depends on an automatic decomposition step, this ablation indicates that a carefully designed prompt largely controls the behavior of the extractor and prevents significant decomposition noise from dominating the uncertainty signal.

## 17 COMPARISON WITH TRADITIONAL SHORT-FORM UQ APPROACHES

We compare GAUSS against two representative short-form uncertainty quantification methods designed for sentence or answer level generation: Semantic Uncertainty [2] and Generating with Confidence [3].

**Why short-form UQ does not directly extend to long-form settings.** Both approaches [2, 3] were originally developed for settings where each query is answered by *short* texts (typically a single sentence), and uncertainty is computed over sets of such short answers. Semantic Uncertainty clusters generations into semantic equivalence classes via NLI-based entailment tests and computes entropy over the resulting "semantic sets." Generating with Confidence constructs a semantic similarity graph

| Model | Dataset | [3] Generating-With-Confidence | | | | [2] Semantic Entropy | | | |
|---|---|---|---|---|---|---|---|---|---|
| | | SC | PC | UCCE | QCCE | SC | PC | UCCE | QCCE |
| Falcon-7B-Instruct | Bios | -0.2946 | -0.2104 | 0.2584 | 0.2294 | -0.1535 | -0.1154 | 0.1863 | 0.1740 |
| | LongFact | -0.2888 | -0.3762 | 0.2448 | 0.2921 | -0.3258 | -0.3731 | 0.1652 | 0.2326 |
| | WildHallu | -0.5122 | -0.5491 | 0.1782 | 0.2227 | -0.5000 | -0.5194 | 0.1592 | 0.1305 |
| LLaMA3-8B-Instruct | Bios | -0.4715 | -0.4517 | 0.1244 | 0.1204 | -0.4307 | -0.4391 | 0.1601 | 0.2410 |
| | LongFact | -0.1355 | -0.2000 | 0.1692 | 0.2431 | -0.1446 | -0.2332 | 0.1574 | 0.2075 |
| | WildHallu | -0.5185 | -0.5162 | 0.1188 | 0.2578 | -0.4578 | -0.5567 | 0.1077 | 0.1933 |
| Qwen2-7B-Instruct | Bios | -0.4387 | -0.4497 | 0.1705 | 0.1739 | -0.3577 | -0.4702 | 0.0888 | 0.1106 |
| | LongFact | -0.2226 | -0.3704 | 0.1888 | 0.2523 | -0.1263 | -0.2825 | 0.0775 | 0.2370 |
| | WildHallu | -0.5978 | -0.6393 | 0.1680 | 0.2504 | -0.4749 | -0.5055 | 0.1666 | 0.2641 |
| Qwen2-57B-Instruct | Bios | -0.4454 | -0.4422 | 0.1171 | 0.1350 | -0.4979 | -0.4994 | 0.1336 | 0.2247 |
| | LongFact | -0.1845 | -0.1750 | 0.2578 | 0.2987 | -0.1231 | -0.1563 | 0.1728 | 0.2192 |
| | WildHallu | -0.5630 | -0.5758 | 0.1526 | 0.2650 | -0.4382 | -0.4722 | 0.1147 | 0.2161 |
| Mistral-7B-Instruct | Bios | -0.4185 | -0.4835 | 0.2045 | 0.1166 | -0.4540 | -0.4619 | 0.1489 | 0.1861 |
| | LongFact | -0.1782 | -0.3037 | 0.1911 | 0.2932 | -0.1644 | -0.2657 | 0.0791 | 0.1678 |
| | WildHallu | -0.5369 | -0.4962 | 0.1596 | 0.2250 | -0.4438 | -0.4944 | 0.1494 | 0.1846 |

Table 12: Short-form UQ baselines adapted to long-form: performance of Generating-with-Confidence [3] and Semantic Uncertainty [2], measured by Spearman (SC), Pearson (PC), and continuous calibration errors (UCCE/QCCE) across models and datasets.

over responses and defines dispersion metrics (e.g., degree-based and Laplacian-eigenvalue scores) on this graph.

In the long-form setting we consider, each generation is a *multi-sentence paragraph* with many entangled factual claims. Applying sentence-level NLI directly to full paragraphs is problematic for two reasons: (i) current NLI models are trained and evaluated primarily on sentence-scale premises and hypotheses and their performance degrades when inputs are long, noisy paragraphs; and (ii) NLI models explicitly lack access to the internal structure of atomic facts within a paragraph, which is precisely what GAUSS is designed to exploit.

**Our long-form adaptations of Semantic Uncertainty and Generating with Confidence.** To nonetheless provide a meaningful comparison, we implement *black-box, paragraph-level* variants of both methods that follow their original spirit but are computationally feasible for long-form outputs.

For Semantic Uncertainty [2], we treat each paragraph as a single response and approximate semantic equivalence between two responses via a semantic-similarity model (rather than exact NLI entailment). We then cluster multiple paragraphs into semantic sets and compute an entropy-based score over the empirical distribution of these sets, analogously to semantic entropy, and use this as the uncertainty measure for each query.

For Generating with Confidence [3], we construct a paragraph-level similarity graph where nodes are entire generations for a given query and edge weights are semantic similarities between paragraphs. On this graph we compute the degree-based uncertainty $U_{\text{Deg}}$ and eigenvalue-based uncertainty $U_{\text{EigV}}$ as defined in [3], and use them as baselines for long-form uncertainty.

**Results.** Table 12 reports Spearman correlation (SC), Pearson correlation (PC), and our continuous calibration metrics (UCCE/QCCE) for both baselines across models and datasets. Overall, we observe that short-form UQ methods, even when adapted to operate on long-form paragraphs, yield noticeably weaker correlations and poorer calibration than GAUSS (from the main text), supporting our hypothesis that modeling the internal semantic graph of atomic facts is important for long-form factuality and uncertainty.