# OpenReview forum: "GAUSS: Graph-Assisted Uncertainty Quantification using Structure and Semantics for Long-Form Generation in LLMs"
_ICLR.cc/2026/Conference — Submitted to ICLR 2026_

### Official Review · Reviewer_9EAH · 2025-10-28

**Soundness:** 3
**Presentation:** 3
**Contribution:** 2
**Rating:** 4
**Confidence:** 3

**Summary:**

In this work, the authors propose a framework GAUSS for measuring uncertainty in long-form LLM outputs through graph-based alignment. The uncertainty is estimated with graph alignment distance between semantic graphs. The authors provide a theoretical justification for GAUSS’s convergence and uncertainty preservation and evaluate it on three benchmarks—bios, LongFact, and WildHallu. Experimental results show that GAUSS significantly improves .

**Strengths:**

1. Strong motivation and interesting idea:
Quantifying uncertainty at the structural level rather than only at token or sentence granularity. The framing is conceptually fresh and highly relevant to factuality and reliability research. The use of semantic graphs coupled with fused Gromov–Wasserstein distances elegantly captures both content overlap and relational coherence among generated facts.

2. Clear methodological exposition.
The pipeline—from atomic fact extraction to graph construction, fusion metric computation, and uncertainty aggregation—is logically structured and well-explained. Pseudo-code and formulas are readable and reproducible.

3. Improved correlation with factuality.
GAUSS achieves notably higher Pearson/Spearman correlations between uncertainty and factual accuracy compared to semantic-only baselines (e.g., SBERT similarity, perplexity-based variance).

**Weaknesses:**

1. Dependence on accurate atomic fact extraction, high cost.
The method’s reliability heavily depends on the quality of factual unit extraction (likely using an entailment or parsing model). Errors in decomposition may propagate and distort both graph structure and distance computation. Fused Gromov–Wasserstein computations are cubic in the number of nodes, making GAUSS computationally heavy for paragraphs with many atomic facts or when using numerous reference generations.

2. Absence of human correlation studies.
The evaluation relies on automatic factuality metrics. Correlating GAUSS scores with human judgments of factual correctness or coherence would strengthen claims of interpretability and practical usefulness.

3. The method relies on a small number of alternative generations (e.g., k = 4–8). While the paper shows concentration theoretically, there’s limited empirical analysis of how uncertainty estimates stabilize with increasing k.

**Questions:**

1. How reliable of the verifier model (Qwen2-32B-Instruct)? The factuality score computation is base on the verifier model.  Do you have human analysis of this verifier model on the three datasets?

2. What are the number of sampled independent paragraph-length response in your experiments? Usually the more , the better. However, if N is large, the computation cost is very high.

---

> ### Author Response · Authors · 2025-11-24
>
> We thank the reviewer for their valuable comments.
>
>
> First, we address the weaknesses section below:
>
> 1. (a) We thank the reviewer for highlighting this concern. Our method adopts the atomic fact extraction procedure commonly used in LUQ, Gen-Binary, Centrality, and related approaches. We fully acknowledge that this step can be susceptible to noise, potentially introducing variability in the resulting uncertainty metrics across all such methods. Importantly, this decomposition is a shared component among GAUSS and the fact-level baselines mentioned, implying that any imperfections in factual unit extraction would, in principle, affect all these methods similarly. **We address this concern in the new section 16 of the updated Appendix.** We also provide a discussion of the same below.
>
>  In our framework, we denote this decomposition model by $M_{\text{atomic}}$ (Qwen2-32B-Instruct in the main experiments) and the corresponding prompt by $p$ (given in Table 6 of the appendix). For a given response paragraph, the atomic fact list produced by $(M_{\text{atomic}}, p)$ is denoted by $AL_{orig}$. To directly address the reviewer’s question, we perform an ablation study that probes the sensitivity of GAUSS to variations in the atomic fact extraction process along two axes: (i) changing the decomposition model $M_{\text{atomic}}$ while keeping the prompt $p$ fixed, and (ii) changing the prompt while keeping $M_{\text{atomic}}$ fixed. Concretely, for (i) we vary the model size (Qwen2-7B,  Qwen2-57B) under the same prompt $p$, and for (ii) we fix $M_{\text{atomic}}$ and consider two prompt variants: removing some atomic facts in the few-shot examples from Table 6, and alternatively, removing all few-shot examples entirely.
>
> Given an alternative atomic fact list $\text{AL}$ for the same paragraph, we quantify its similarity to $\text{AL}_{\text{orig}}$ using two complementary metrics:
>
> **LLM-based overlap score**. For each atomic fact $f \in \text{AL}$, we query an LLM (Qwen2.5-32B model) to decide whether $f$ is semantically present in $\text{AL}-{\text{orig}}$ (outputting 1 if supported and 0 otherwise). The overlap score between $\text{AL}$ and $\text{AL}-{\text{orig}}$ is the mean of these binary labels over all $f \in \text{AL}$. We report the average overlap score across all responses in a dataset; higher values indicate that the two decompositions capture essentially the same factual content.
>
>
> **GAUSS graph distance**: Each atomic fact list (both $\text{AL}$ and $\text{AL}-{\text{orig}}$) induces a semantic graph under the GAUSS construction. We compute the GAUSS alignment distance $D(\text{AL}, \text{AL}-{\text{orig}})$ between the two induced graphs and report the mean distance across all responses; lower values indicate more similar graph structure and hence a more stable decomposition.
>
>
> Empirically, we find that changing the prompt (while holding $M_{\text{atomic}}$ fixed) leads to noticeably lower overlap scores and higher graph distances, reflecting the fact that very different prompting can change how aggressively the model splits or merges facts. In contrast, when we change the model size (Qwen2-7B / 57B) under a fixed, well-specified prompt $p$, the overlap scores remain high and the GAUSS graph distances remain small, indicating that the resulting atomic fact lists are very close both semantically and structurally.
> These results suggest that, in practice, once a stable prompt $p$ is chosen (such as the one we provide in Table 6), the choice of $M_{\text{atomic}}$ has a modest effect on the induced semantic graphs and the resulting GAUSS uncertainty scores. While GAUSS, like all fact-level methods, necessarily depends on an automatic decomposition step, this ablation indicates that a carefully designed single prompt largely controls the behavior of the extractor and prevents significant decomposition noise from dominating the uncertainty signal.
>
> | Setting                         | \(M_{\text{atomic}}\)        | Prompt                   | Overlap ↑ | Mean Graph Dist. ↓ |
> |---------------------------------|------------------------------|--------------------------|-----------|---------------------|
> | Varying model, fixed \(p\)      | Qwen2-7B-Instruct            | \(p\) (orig.)            | 0.9619    | 0.1288              |
> | Varying model, fixed \(p\)      | Qwen2.5-57B-Instruct         | \(p\) (orig.)            | 0.9823    | 0.0412              |
> | Varying prompt, fixed Qwen2.5-32B | Qwen2.5-32B-Instruct       | \(p\) (few-shot pruned)  | 0.9591    | 0.1786              |
> | Varying prompt, fixed Qwen2.5-32B | Qwen2.5-32B-Instruct       | \(p\) (no few-shot)      | 0.9037    | 0.1960              |

---

> ### Author Response · Authors · 2025-11-24
>
> 1.  (b) We also want to briefly clarify the computational complexity of GAUSS:
> (i) First, the GAUSS uncertainty metric is : $U(q) = \frac{1}{N-1} \sum_{r \neq a} D_{\alpha}(G_a, G_r)$ . Each term in the summation is computed fully parallely. Thus, the uncertainty computation can scale well with the number of sampled paragraphs. We show the runtime of the entire GAUSS procedure from atomic fact decomposition, to uncertainty computation in Section 9 of the Appendix.
> (ii) Next, the computation of the graph distance (FGW) metric $D_{\alpha}(G_a, G_r)$ although being $O(n_a^2 \cdot n_r^2)$, is fast to compute for $n_a, n_r \approx 50$. This is explained by the fact that the underlying components in the distance metric including $M^r[i,j]$ and $C[i,k,j,l]$ are precomputed.
>  We also show practical runtimes of the computation of $D_{\alpha}(G_a, G_r)$ in **Section 15 of the updated Appendix**.
>
>
> 2. We appreciate this suggestion and agree that human studies are an important direction for future work. In this paper, we chose to rely on factuality metric verification techniques (previously used in the literature [1]),  such as SAFE-style pipelines where relevant scraped web-pages are presented to the verifier model along with the fact in question to classify whether it is supported by the facts scraped from the web. As a result, we believe our “veracity” targets reflect human factual judgments indirectly, and we evaluate GAUSS across multiple datasets and models against these standardized metrics. That said, we fully acknowledge that a dedicated human study comparing GAUSS scores with direct human ratings of factuality and coherence would further strengthen the interpretability and practical usefulness of the method, and we view this as a valuable direction for future work beyond the current submission.
>
> 3. We would like to clarify that in all our experiments we actually use a larger  N=20 generations per query, not k=4-8; we have made this explicit earlier in Section 5 of the updated manuscript to avoid  confusion.
>
> Furthermore, we show empirical experimentation on the meaning of the convergence result in Theorem 4.2 in **Section 7 of the Appendix**.

---

> ### Author Response · Authors · 2025-11-24
>
> Next, we address the questions below:
>
> 1.  We agree that the reliability of the verifier is crucial, since our paragraph-level factuality labels are derived from it. In our setup we do not use the raw Qwen2-32B output as a bare classifier: we use it within our SAFE-style framework [1], where each atomic fact is checked open-book against web-retrieved evidence and the model is only asked to judge whether the fact is supported, refuted, or unverifiable given those snippets. This makes the verifier behave much more like a human annotator with access to external sources, and mitigates many of the hallucination issues that arise in closed-book evaluation. This is in accordance with the compared baselines (LUQ, Gen-binary etc).
>
>
> 2. We use **N= 20** i.e. we sample 20 paragraph level responses from the model and compute the uncertainty metric of GAUSS. We agree that larger N is better. However, since each graph distance computation can be done fully independently and  parallely from the other,  we see that our approach does not incur significant runtime overheads with larger N.
> The GAUSS uncertainty metric is : $U(q) = \frac{1}{N-1} \sum_{r \neq a} D_{\alpha}(G_a, G_r)$ . Each term in the summation is computed fully parallely. To make this more evident, we compare runtimes of our method in Section 9 and Section 15 of the updated Appendix.
>
> References:
> [1] Jerry Wei et al. “Long-form factuality in large language models”. In: arXiv preprint arXiv:2403.18802 (Mar. 2024). URL: https://arxiv.org/abs/2403.18802.
>
>
> We hope we clarify most of your concerns. We will be very happy to answer any other questions you may have.

---

> ### Author Response · Authors · 2025-12-03
> **For the AC: Summary of Rebuttal with Reviewer 9EAH**
>
> We briefly postulate points constituting  the reviewer’s main concerns followed by a summary of our response in the rebuttal process until now.
>
>
> 1. **Reviewer’s Concern: Auxiliary LLM for atomic fact decomposition introduces noise in the UQ pipeline.**
>
> A summary of our response is stated below:
>
> - Atomic fact decomposition is very necessary to provide a representation of long form paragraphs.
> - Moreover, this atomic fact decomposition step is shared across all fact-level long form UQ  baselines existent in the literature (LUQ, Gen-Binary, Centrality)
> -  However, we agree with the reviewer about the noise introduced by the atomic fact extraction process.  In response to that , we added Appendix Sec. 16 with a systematic sensitivity study over (i) different decomposition models (Qwen2-7B / 32B / 57B) under a fixed prompt and (ii) different prompts under a fixed model. Using overlap scores and GAUSS graph distances, we show that with a stable prompt the decomposition is highly stable and thereby does not affect the downstream UQ measure (across GAUSS and all other compared baselines too).
>
> 2.  **Reviewer’s Question: Computational cost of GAUSS**
>
>
> A summary of our response is below:
> - Clarified complexity and practical runtimes:
>
>  (i) GAUSS uncertainty uses $ U(q) = \frac{1}{N-1} \sum_{r \neq a} D_\alpha(G_a, G_r) $, with all pairwise distances computed in parallel.
>
>
> (ii) FGW distance $ D_\alpha(G_a, G_r) $ is $ O(n_a^2 n_r^2) $, but is fast in practice for $ n_a, n_r \approx 50 $ because of precomputed $ M^r $ and structural costs.
>
>
> (iii) We add  runtime tables and discussion in the updated Appendix Sec. 9 and Sec. 15 to show end-to-end wall-clock times (fact extraction + GAUSS).
>
>
> 3.  **Reviewer’s Question: Absence of human correlation studies**
> A summary of our rebuttal is below:
>
>
> - We explain that we follow a SAFE-style factuality pipeline (as in Wei et al. 2024), where Qwen2-32B is used open-book with web evidence to label atomic facts as supported/refuted/unverifiable - so our “veracity” targets already proxy human factual judgments.
>
>
> - We also acknowledge that direct human rating studies of GAUSS vs. human-perceived factuality/coherence are valuable and explicitly flagged this as future work beyond the scope of the current submission.
>
>
> 4. **Reviewer’s Question: Number of generations (N) and stabilization with (N)**
>
> A summary of our response is below:
>
>
> - We clarified that in all experiments we use (N = 20) paragraph-level generations, not (k = 4{-}8); this is now made explicit in Sec. 5 of the main mansucript.
>
>
> - We emphasize that pairwise distances are massively parallel, so larger (N) does not incur prohibitive cost.
>
>
> - We point the reviewer to Appendix Sec. 7 with empirical analysis illustrating how uncertainty estimates behave in light of the convergence result (Theorem 4.2).
>
>
> 5. **Reviewer’s Concern: Reliability of the verifier (Qwen2-32B-Instruct)**
> A summary of our response is as below:
>
>
> -  We clarify that the verifier is not used as a bare classifier: instead, it operates in a SAFE-style open-book setting with retrieved web evidence per fact, and only has to decide “supported / refuted / unverifiable”.
>
>
> - This setup mitigates hallucination, **aligns with prior long-form factuality work**, and is shared with baselines like LUQ / Gen-Binary, ensuring a fair comparison.

---

### Official Review · Reviewer_yDmm · 2025-10-31

**Soundness:** 4
**Presentation:** 3
**Contribution:** 4
**Rating:** 6
**Confidence:** 3

**Summary:**

The authors derive better calibrated uncertainty quantification scores by modeling facts as semantic graphs

**Strengths:**

The motivation is clear and natural. The experimental set up is clear, and their method outperforms prior methods in most cases.

In general, I am not very familiar with this space of papers that try to come up with new and better uncertainty scores for LLMs. While I think this is a nice contribution, there could be prior work outside of those that the authors directly compare to that I do not know about.

**Weaknesses:**

My weaknesses are minor
- The writing could be improved. The writing feels a little cramped and clumsy, which sometimes made it difficult to parse through everything.
- The general idea of framing facts as graphs for LLM uncertainty is not new (as stated by the authors)
- There are cases where GAUSS underperforms prior methods

**Questions:**

1. Could you also report expected calibration error (let's say for 10 bins) and Brier score? I find that those metrics are easier to interpret, and I think certain groups are researchers for uncertainty quantification who are more used to those metrics would appreciate seeing those results as well.
2. You mention in the limitations that your method is computationally expensive. Could you quantify and compare the computational costs for your experiments (perhaps in terms of just compute time for your machine)?

Suggestion:
1. use 3 instead of 4 decimal places for Table 1
2. Citing [1] - similar in that they also consider graph structures of LLM outputs (reasoning chains) for uncertainty quantification (post-hoc, conformal prediction)

[1] Rubin-Toles, Maxon, et al. "Conformal Language Model Reasoning with Coherent Factuality." The Thirteenth International Conference on Learning Representations.

---

> ### Author Response · Authors · 2025-11-24
>
> We dearly thank the reviewer for their thoughtful comments and suggestions.
>
> We address the questions enlisted by the reviewer below:
>
> 1.  Classical expected calibration error (ECE) and Brier score are indeed standard and widely used in the uncertainty quantification community, especially in settings with binary ground-truth labels and probabilistic predictions. In our work, however, the ground-truth “veracity” is inherently continuous. It is defined as the fraction of supported atomic facts in a paragraph, while our methods output continuous uncertainty scores. For this continuous-continuous setting, we therefore focused on UCCE and QCCE, which are specifically designed to assess calibration when both the target and the predictions lie in a continuous range, and are more faithful to the underlying semantics of our task.
>
> That said, we fully agree that ECE and Brier score are intuitive and familiar to many UQ researchers. To accommodate this suggestion, in the revised version we additionally construct a binary veracity label by thresholding paragraph veracity at 0.5 (paragraphs with factuality > 0.5 are treated as “correct”, others as “incorrect”), and then compute standard Brier score and ECE with 10 bins on this binarized label space. The corresponding results for GAUSS and all baselines are now reported in **Section 14 of the updated  Appendix.**
>
> 2.  We report the computational complexity of GAUSS and also the runtimes of the baseline method in **Section 9 of the Appendix.**
>
> 3.  We will make the correction to the decimal places in the final updated manuscript.
>
> 4.  We have included the work [1] in the related works section in the updated manuscript.
>
> Specifically, [1] exploit graph structure in LLM outputs across reasoning steps, representing chain-of-thought solutions as deducibility graphs and applying split conformal prediction to obtain coverage guarantees for “coherent factuality” in reasoning tasks. Their focus is thus on conformal filtering over reasoning graphs for mathematical problems, whereas GAUSS targets long-form factual generations and quantifies uncertainty via optimal-transport alignment between semantic graphs of atomic facts.
>
> We hope this answers your concerns. We will be very happy to answer any other questions you may have.

---

> > ### Comment · Reviewer_yDmm · 2025-11-25
> >
> > Thank you for answering my questions and addressing my suggestions.

---

> ### Author Response · Authors · 2025-12-03
> **For the AC: Summary of Rebuttal with Reviewer yDmm**
>
> We briefly postulate points constituting  the reviewer’s main concerns followed by a summary of our response in the rebuttal process until now.
>
>
> 1. **Reviewer’s Question: Inclusion of ECE & Brier score in calibration results.**
> A summary of our response is below:
> - We explained that our setting is continuous–continuous (veracity = fraction of supported facts; GAUSS outputs continuous uncertainty), so we primarily use UCCE/QCCE, which are designed for this regime.
> - To directly address the reviewer’s request, we binarize paragraph veracity at 0.5 (“correct” vs “incorrect”) and report standard 10-bin ECE and Brier score for GAUSS and all baselines in Section 14 of the updated Appendix.
>
> 2. **Reviewer’s Question: Reporting of runtime**
> A summary of our response is below.
> - We direct the reviewer to the reporting of the complexity and practical runtimes of GAUSS and baselines in Section 9 of the Appendix.
>
> 3. **Reviewer’s Question: Minor corrections & related work citing**
> A summary of our response is below.
> - We assure to fix the decimal-places to 3 in the new manuscript.
>
>
> - We added the suggested related work [1] to the related-work section and clarified how that graph-over-reasoning-steps line of work differs from GAUSS, which targets long-form factual paragraphs and uses OT over semantic graphs of atomic facts rather than conformal filtering over reasoning traces.

---

### Official Review · Reviewer_GxGz · 2025-10-31

**Soundness:** 2
**Presentation:** 2
**Contribution:** 2
**Rating:** 4
**Confidence:** 3

**Summary:**

In this paper, the author introduces an Uncertainty quantification method: Graph-Assisted Uncertainty Quantification Using Structure and Semantics (GAUSS), it is a framework for measuring uncertainty in long-form LLM generation by representing each paragraph as a semantic graph that encodes both factual content and inter-fact structure. Uncertainty is computed as the fused Gromov–Wasserstein distance between an anchor and reference graphs; this way, the author shows that it can capture both semantic and structural variation. The author tries to include the theoretical guarantees on robustness and convergence of the method, and by experiment, it achieves state-of-the-art correlation and calibration with factual accuracy across multiple datasets and models.

**Strengths:**

1. The paper proposes a framework, GAUSS, that quantifies uncertainty in long-form LLM generation using semantic graphs.
2. It integrates both semantic meaning and inter-fact structural relations through a fused Gromov–Wasserstein distance, which could provide an angle to measure paragraph-level variability across generations.


3. The experimental results are thorough and convincing, covering multiple datasets and models, and demonstrate that GAUSS achieves stronger correlation with factual accuracy and better calibration than existing baselines such as LUQ and Gen-Binary.

**Weaknesses:**

1. GAUSS relies on an auxiliary LLM for fact decomposition, which may introduce extra uncertainty unrelated to the base model’s generation, and the paper lacks analysis of its sensitivity to different decomposition models or prompts.


2. The constructed semantic graph seems fully connected, with edges defined by pairwise semantic similarities, raising doubts about whether it captures structural characteristics beyond aggregated semantics.


3. The work lacks the majority of the baselines that are classic in the UQ for the LLM domain, such as semantic uncertainty, degree-based sparsity / confidence measurement, it would be good to see more standard evaluation to emphasize the contribution of the proposed method.

**Questions:**

1. GAUSS depends on an auxiliary LLM $M_{atomic}$ to decompose paragraphs into atomic facts before computing graph-based uncertainty. Since this step alters the original outputs, how can authors be sure that GAUSS measures the generation uncertainty of the base LLM rather than the decomposition uncertainty introduced by $M_{atomic}$ ? Have the authors tested the sensitivity of results to different decomposition models or prompts?


2. The paper states that GAUSS constructs a semantic graph where nodes are atomic facts and edges encode pairwise semantic relationships. Is this graph fully connected (i.e., every node pair linked by a weighted edge)? If so, could the resulting structure simply reflect pairwise semantic similarities rather than the structural characteristics? In that case, how does GAUSS ensure that its “structural” component captures more than just aggregated semantic similarity?


3. The paper claims that GAUSS captures “structural uncertainty” by modeling paragraphs as semantic graphs and measuring alignment via the fused Gromov–Wasserstein distance. However, since the edges are defined through pairwise semantic similarities between atomic facts, the resulting graph structure seems derivative of semantics. Could the authors clarify what constitutes the structural representation in GAUSS and what it actually represents?


4. The paper uses various terms and can be misleading sometimes, e.g., could the author confirm if the factuality score holds the same meaning as the correctness? It is suggested to use one consistently.


5. If the description of $C_i​$ as in line 253 suggests a fully connected graph, however, the presentation in Figure 1 (Semantic Graph Creation from Atomic Facts), and Figure 3 all illustrate partially connected edges, which is confusing. Could the authors clarify the actual connectivity of the nodes within the semantic graph?


6. Literature and related work:


- There seems like a very similar existing work [1], which also leverages the structural and semantic meaning to quantify the uncertainty of the LLMs. Could the author discuss the difference from this related work? Especially, this work applies the structural measure for the reasoning topology. How does the author’s work differ?


[1] Da, Longchao, et al. “Understanding the Uncertainty of LLM Explanations: A Perspective Based on Reasoning Topology.” Proceedings of the Second Conference on Language Modeling (COLM 2025).


- There is no comparison from any of the work on semantic uncertainty [2] or uncertainty quantification methods from work [3], which is a widely acknowledged baseline method group in the uncertainty quantification of LLMs, raising concerns about the actual performance of the proposed method.


[2] Kuhn, Lorenz, Yarin Gal, and Sebastian Farquhar. "Semantic uncertainty: Linguistic invariances for uncertainty estimation in natural language generation." arXiv preprint arXiv:2302.09664 (2023).


[3] Lin, Zhen, Shubhendu Trivedi, and Jimeng Sun. "Generating with confidence: Uncertainty quantification for black-box large language models." arXiv preprint arXiv:2305.19187 (2023).

---

> ### Author Response · Authors · 2025-11-24
>
> We thank the reviewer for their thoughtful comments and suggestions.
>
> We first address the weaknesses pointed out :
>
> 1.  We thank the reviewer for highlighting this concern. Our method adopts the atomic fact extraction procedure commonly used in LUQ, Gen-Binary, Centrality, and related approaches. We fully acknowledge that this step can be susceptible to noise, potentially introducing variability in the resulting uncertainty metrics across all such methods. Importantly, this decomposition is a shared component among GAUSS and the fact-level baselines mentioned, implying that any imperfections in factual unit extraction would, in principle, affect all these methods similarly. **We address this concern in the new section 16 of the updated appendix.** We also provide a discussion of the same below.
>
>  In our framework, we denote this decomposition model by $M_{\text{atomic}}$ (Qwen2-32B-Instruct in the main experiments) and the corresponding prompt by $p$ (given in Table 6 of the appendix). For a given response paragraph, the atomic fact list produced by $(M_{\text{atomic}}, p)$ is denoted by $AL_{orig}$. To directly address the reviewer’s question, we perform an ablation study that probes the sensitivity of GAUSS to variations in the atomic fact extraction process along two axes: (i) changing the decomposition model $M_{\text{atomic}}$ while keeping the prompt $p$ fixed, and (ii) changing the prompt while keeping $M_{\text{atomic}}$ fixed. Concretely, for (i) we vary the model size (Qwen2-7B,  Qwen2-57B) under the same prompt $p$, and for (ii) we fix $M_{\text{atomic}}$ and consider two prompt variants: removing some atomic facts in the few-shot examples from Table 6, and alternatively, removing all few-shot examples entirely.
>
> Given an alternative atomic fact list $\text{AL}$ for the same paragraph, we quantify its similarity to $\text{AL}_{\text{orig}}$ using two complementary metrics:
>
> **LLM-based overlap score**. For each atomic fact $f \in \text{AL}$, we query an LLM (Qwen2.5-32B model) to decide whether $f$ is semantically present in $\text{AL}-{\text{orig}}$ (outputting 1 if supported and 0 otherwise). The overlap score between $\text{AL}$ and $\text{AL}-{\text{orig}}$ is the mean of these binary labels over all $f \in \text{AL}$. We report the average overlap score across all responses in a dataset; higher values indicate that the two decompositions capture essentially the same factual content.
>
>
> **GAUSS graph distance**: Each atomic fact list (both $\text{AL}$ and $\text{AL}-{\text{orig}}$) induces a semantic graph under the GAUSS construction. We compute the GAUSS alignment distance $D(\text{AL}, \text{AL}-{\text{orig}})$ between the two induced graphs and report the mean distance across all responses; lower values indicate more similar graph structure and hence a more stable decomposition.
>
>
> Empirically, we find that changing the prompt (while holding $M_{\text{atomic}}$ fixed) leads to noticeably lower overlap scores and higher graph distances, reflecting the fact that very different prompting can change how aggressively the model splits or merges facts. In contrast, when we change the model size (Qwen2-7B / 57B) under a fixed, well-specified prompt $p$, the overlap scores remain high and the GAUSS graph distances remain small, indicating that the resulting atomic fact lists are very close both semantically and structurally.
> These results suggest that, in practice, once a stable prompt $p$ is chosen (such as the one we provide in Table 6), the choice of $M_{\text{atomic}}$ has a modest effect on the induced semantic graphs and the resulting GAUSS uncertainty scores. While GAUSS, like all fact-level methods, necessarily depends on an automatic decomposition step, this ablation indicates that a carefully designed single prompt largely controls the behavior of the extractor and prevents significant decomposition noise from dominating the uncertainty signal.
>
> | Setting                         | \(M_{\text{atomic}}\)        | Prompt                   | Overlap ↑ | Mean Graph Dist. ↓ |
> |---------------------------------|------------------------------|--------------------------|-----------|---------------------|
> | Varying model, fixed \(p\)      | Qwen2-7B-Instruct            | \(p\) (orig.)            | 0.9619    | 0.1288              |
> | Varying model, fixed \(p\)      | Qwen2.5-57B-Instruct         | \(p\) (orig.)            | 0.9823    | 0.0412              |
> | Varying prompt, fixed Qwen2.5-32B | Qwen2.5-32B-Instruct       | \(p\) (few-shot pruned)  | 0.9591    | 0.1786              |
> | Varying prompt, fixed Qwen2.5-32B | Qwen2.5-32B-Instruct       | \(p\) (no few-shot)      | 0.9037    | 0.1960              |

---

> ### Author Response · Authors · 2025-11-24
>
> 2.  We appreciate the reviewer’s careful question about what is truly “structural” in GAUSS.
>
> We answer this concern in two parts:
>
> a) Conceptually, GAUSS operates on **two coupled levels**. At the first order, the semantic cost term $ M^r $ aligns individual atomic facts across two paragraphs by comparing their sentence embeddings and factual support, i.e., “does fact $i$ in paragraph A match fact $j$ in paragraph B?”. At the second order, the structural cost tensor $ L^r $ aligns patterns of relationships between facts: when the transport plan matches $i \leftrightarrow j$ and $k \leftrightarrow \ell$, the term $ L^r $ penalizes discrepancies between the intra-paragraph distances $ C_a(i,k) $ and $ C_r(j,\ell) $. In other words, GAUSS does not only ask whether individual facts are semantically similar, but also whether the *way they co-occur* within each paragraph is consistent across generations. This uses **local pairwise semantic similarities as building blocks**, but the fused Gromov–Wasserstein alignment aggregates them at the level of **global co-dependence** of facts in the paragraph, which is precisely what we refer to as structure.
>
> b)  To disentangle GAUSS’ captured  structure from mere “aggregated semantic similarity”, we also perform an empirical control experiment. For the GAUSS framework to use aggregated semantic similarity instead of its structural term, we replace the full structural cost tensor $ L^r $ with a **per-node aggregated similarity score**: for each fact $ i $ in a paragraph we compute
>
> $ s(i) = \frac{1}{|V|} \sum_{k} \text{sim}(f_i, f_k), $
>
> i.e., the mean pairwise semantic similarity of $i$ to all other facts in the same paragraph. We then define an alternative alignment that uses the same semantic term $M^r$ as GAUSS, but augments node matching only with these scalar summaries $ s(i) $, rather than the full pairwise structure matrix $ L^r $.
> Concretely, we induce a cost term for matching $ i $ and $j$, $ Cost(i,j) $ as:
> $ Cost(i,j) = M^r[i,j] + |s(i) - s(j)| $
>
>  Intuitively, this “aggregated semantic similarity” variant, captures how *central* or *generic* each fact is on average, but discards the detailed pattern of which facts are close to which others. When we compare this variant to GAUSS on correlation with veracity labels and on the induced distance distributions, we observe that the aggregated-similarity variant yields weaker correlations with veracity than GAUSS. This empirical gap, together with the two-level design of $ M^r $ and $ L^r $, supports our claim that GAUSS’s structural component is not just re-using semantics in aggregate, but is genuinely leveraging the *geometry of fact relationships* - i.e., the global paragraph structure beyond what can be captured by simple aggregations of pairwise similarities.
>
> We provide the results for semantic pairwise aggregation and GAUSS below for the bios dataset with the qwen2-7b-instruct model’s responses.
>
> | Method              | SC       | PC       | UCCE   | QCCE   |
> |---------------------|----------|----------|--------|--------|
> | Semantic Aggregation| -0.4398  | -0.3789  | 0.1800 | 0.2180 |
> | GAUSS               | -0.6915  | -0.7114  | 0.1606 | 0.1505 |

---

> ### Author Response · Authors · 2025-11-24
>
> 3.  We appreciate the reviewer’s suggestion regarding broader baseline comparisons. Since our work focuses specifically on uncertainty quantification in long-form generation, we have made all efforts to include relevant baselines from this domain (LUQ, Gen-Binary, Centrality, Dis-Rating, Dis-Single etc).
>
> Classical short-form UQ approaches, such as semantic entropy [2], operate by clustering generations into semantic equivalence classes via bi-directional entailment and computing entropy over these clusters. This paradigm has been extended by [3], who propose a range of semantic-similarity-based metrics, including measures based on the number of semantic sets  and degree-based sparsity/confidence.
>
> While these methods are effective for sentence-level responses, they are less suited for long-form content, as entailment models struggle to capture nuanced meaning and logical relationships at the paragraph level. **Nonetheless, in line with the reviewer’s recommendation, we have incorporated a comprehensive comparison to these traditional short-form UQ methods in Section 17 of the Appendix to contextualize our contributions more fully.**

---

> ### Author Response · Authors · 2025-11-24
>
> We next address the Questions posed by the reviewer:
> 1.  We address questions on the noise induced by atomic fact extraction in Section 16 of the Appendix and also in point 1 of Weaknesses above.
>
> 2.  We provide a justification of how GAUSS ensures that its “structural” component captures more than just aggregated semantic similarity in point 2 of the Weaknesses section above.
>
> 3. We define  the structural component of GAUSS in point 2 of the Weaknesses section above.
>
> 4. We apologise for the confusion. Factuality and correctness are used interchangeably. However, we will correct this in the text to adhere to a single term usage.
>
> 5. The connectivity of the semantic graph is a soft connectivity where each pairwise interaction is equal to the semantic cosine similarity. We do not explicitly indicate the full-connectivity in the graphs for illustration purposes. However, GAUSS is equipped automatically in handling graphs that are not fully connected (zero cosine similarity case).
>
> 6. We answer this in two parts:
>
>
> (a)  [1] and our work target related but ultimately complementary regimes of UQ. Topo-UQ is designed for reasoning paths: it explicitly elicits chain-of-thought style explanations, decomposes them into knowledge-subquestion / sub-answer pairs, and then asks the LLM itself to wire these into a directed reasoning topology whose nodes and edges are meant to reflect causal or logical dependencies between reasoning steps; uncertainty is then quantified via a tailored graph-edit-distance variant over these explanation graphs. In contrast, GAUSS is built for long-form factual generations, where the model may not expose an explicit reasoning trace at all: each paragraph is first decomposed into atomic facts and represented as a dense, weighted semantic graph whose nodes are factual statements and whose edge weights are soft pairwise semantic affinities; we then compare multiple such graphs via a fused Gromov-Wasserstein optimal-transport distance that jointly penalizes mismatches in node semantics and in the induced geometric structure of inter-fact relationships. This OT-based alignment allows us to derive Lipschitz and exponential-convergence guarantees for the resulting uncertainty score, and to operate post-hoc on arbitrary long-form outputs without requiring any special explanation-elicitation protocol, making our setting, graph construction, and distance notion fundamentally different from the reasoning-topology and graph-edit-distance framework of Topo-UQ. We provide a reference to [1] in the updated manuscript.
>
> (b) We address comparisons to traditional LLM output UQ methods like [2], [3] in point 3 of the Weaknesses section above and also in Section 17 of the updated Appendix.
>
>
>
> We hope this answers your concerns. We will be very happy to answer any other questions you may have.

---

> ### Author Response · Authors · 2025-12-03
> **For the AC : Summary of Rebuttal with Reviewer GxGz**
>
> We briefly postulate points constituting  the reviewer’s main concerns followed by a summary of our response in the rebuttal process until now.
>
> 1. **Reviewer’s Concern: Auxiliary LLM for atomic fact decomposition introduces noise in the UQ pipeline.**
>
> A summary of our response is stated below:
>
> - Atomic fact decomposition is very necessary to provide a representation of long form paragraphs.
> - Moreover, this atomic fact decomposition step is shared across all fact-level long form UQ  baselines existent in the literature (LUQ, Gen-Binary, Centrality)
> -  However, we agree with the reviewer about the noise introduced by the atomic fact extraction process.  In response to that , we added Appendix Sec. 16 with a systematic sensitivity study over (i) different decomposition models (Qwen2-7B / 32B / 57B) under a fixed prompt and (ii) different prompts under a fixed model. Using overlap scores and GAUSS graph distances, we show that with a stable prompt the decomposition is highly stable and thereby does not affect the downstream UQ measure (across GAUSS and all other compared baselines too).
>
> 2. **Reviewer’s Concern: Clarification on how GAUSS captures structure and its connection with aggregated semantic similarity.**
>
> A summary of our response is stated below:
> - We clarify GAUSS’s two-level design (semantic term $ M^r $ + structural term $ L^r $) that jointly capture semantic similarity and the co-dependence of atomic facts between two paragraphs. To emphasize why GAUSS captures more information than aggregated semantic similarity, we  add a control variant that uses only aggregated semantic similarity (per-node mean similarity) instead of the full structural tensor in the rebuttal above. GAUSS significantly outperforms this variant on correlation and calibration, showing the importance of the structural component in capturing the  global geometry of fact relationships.
> 3. **Comparison with classic short-form UQ baselines**
>
> A summary of our response is below:
> - In the main manuscript, we perform experimentation with respect to all long form UQ approaches in the literature.
> - We add a new  Section 17 in the  Appendix, discussing the shortcomings of classical short form UQ approaches for long form UQ. In line with the reviewer’s concern, we also include experimentations of how classical UQ methods may fare in long form settings in the same Section 17 of the updated Appendix.
>
> 4. **Question on Graph connectivity and illustrations**
>
> A summary of our response is below:
> - We clarify that the semantic graph is conceptually fully connected (all pairs have cosine-similarity edge weights, zeros act as missing edges). Figures only show a subset of edges for readability; GAUSS naturally handles sparse/zero edges.

---

### Author Response · Authors · 2025-12-03

Dear AC,

To the best of our knowledge, we have addressed all concerns raised by the reviewers. We have also included a brief summary of our responses to each reviewer (under "For the AC : Summary of Rebuttal with Reviewer {reviewer}" official comment), highlighting their main questions/concerns and how we have responded to them.

We sincerely thank the reviewers and the area chairs for their time and thoughtful feedback.

Best regards,

The Authors

---

### Meta-Review · Area_Chair_hPiT · 2026-01-07

**Summary:**

This paper proposes GAUSS, a graph-based framework for uncertainty quantification in long-form LLM generation that represents paragraphs as semantic graphs over atomic facts and measures uncertainty. Reviewers agree that the problem setting is timely and important, and that the method is technically well constructed, with generally improved correlation and calibration relative to several existing long-form uncertainty baselines.

However, the reviews consistently raise concerns regarding the method's heavy reliance on automatic atomic fact decomposition and verification pipelines, the extent to which the proposed graph structure captures genuinely new structural signals beyond aggregated semantic similarity, the high computational cost of graph alignment, and the absence of direct human correlation studies. While the rebuttal provides clarifications, ablations, and additional metrics, these responses mitigate but do not fully resolve the core issues.

**Reviewer Concerns:**

**Concerns Addressed:**
* The authors provide ablations across decomposition models and prompts, showing relative stability under a carefully fixed prompt.
* Additional conceptual explanation and a control experiment distinguishing GAUSS from aggregated semantic similarity strengthen the technical justification.
* Requested metrics and runtime analyses were added in the appendix, addressing reviewer requests.
* Missing related work on graph-based and conformal uncertainty was added and appropriately discussed.

**Concerns Still Outstanding:**
* GAUSS still relies heavily on LLM-based fact extraction and verification, and no direct human correlation study is provided to validate that GAUSS uncertainty aligns with human perceptions of factual reliability or coherence.
* Despite discussion of parallelization, fused Gromov–Wasserstein alignment remains computationally expensive, raising practical concerns for longer paragraphs or large-scale deployment.
* Several important validations remain appendix-only, and the lack of human evaluation limits the strength of claims about interpretability and real-world usefulness.
* While technically solid, the core idea of graph-based uncertainty modeling is incremental relative to prior work, and the practical gains over existing approaches may be narrower than implied.

**Reviewer Scores:**

Reviewer GxGz: 4 -> 4
* Core concerns about decomposition noise, structural interpretation, and baseline coverage are clarified but not resolved.

Reviewer yDmm: 6 -> 6
* Added metrics and citations address requests but are incremental rather than score-changing.

Reviewer 9EAH: 4 -> 4
* Concerns about computational cost, reliance on automatic verification, and lack of human studies persist.

**Overall:**
GAUSS is a carefully engineered and technically sound framework with promising empirical behavior. However, the remaining concerns regarding reliance on automatic pipelines, lack of direct human validation, limited practical scalability, and incremental conceptual novelty outweigh the strengths. While the paper could become strong with additional validation and scope expansion, it does not currently stand clearly above the acceptance bar at this venue, and I therefore recommend rejection.

---

### Decision · Program_Chairs · 2026-01-26

Reject